# DCI-ES: An Extended Disentanglement Framework with Connections to Identifiability

**Cian Eastwood**[*,1,2], **Andrei Liviu Nicolicioiu**[*,1], **Julius von Kügelgen**[*,1,3],
**Armin Kekić**[1], **Frederik Träuble**[1], **Andrea Dittadi**[1,4], **and Bernhard Schölkopf**[1]

[1]Max Planck Institute for Intelligent Systems, Tübingen, Germany
[2]School of Informatics, University of Edinburgh
[3]Department of Engineering, University of Cambridge
[4]Technical University of Denmark

## Abstract

In representation learning, a common approach is to seek representations which disentangle the underlying factors of variation. Eastwood & Williams (2018) proposed three metrics for quantifying the quality of such disentangled representations: disentanglement (D), completeness (C) and informativeness (I). In this work, we first connect this DCI framework to two common notions of linear and nonlinear identifiability, thereby establishing a formal link between disentanglement and the closely-related field of independent component analysis. We then propose an extended DCI-ES framework with two new measures of representation quality—*explicitness* (E) and *size* (S)—and point out how D and C can be computed for black-box predictors. Our main idea is that the *functional capacity required to use a representation* is an important but thus-far neglected aspect of representation quality, which we quantify using explicitness or *ease-of-use* (E). We illustrate the relevance of our extensions on the `MPI3D` and `Cars3D` datasets.

## 1 Introduction

A primary goal of representation learning is to learn representations $r(x)$ of complex data $x$ that "make it easier to extract useful information when building classifiers or other predictors" (Bengio et al., 2013). *Disentangled* representations, which aim to recover and separate (or, more formally, *identify*) the underlying factors of variation $z$ that generate the data as $x = g(z)$, are a promising step in this direction. In particular, it has been argued that such representations are not only interpretable (Kulkarni et al., 2015; Chen et al., 2016) but also make it easier to extract useful information for downstream tasks by recombining previously-learnt factors in novel ways (Lake et al., 2017).

While there is no single, widely-accepted definition, many evaluation protocols have been proposed to capture different notions of disentanglement based on the relationship between the learnt representation or *code* $c = r(x)$ and the ground-truth data-generative factors $z$ (Higgins et al., 2017; Eastwood & Williams, 2018; Ridgeway & Mozer, 2018; Kim & Mnih, 2018; Chen et al., 2018; Suter et al., 2019; Shu et al., 2020). In particular, the metrics of Eastwood & Williams (2018)—*disentanglement* (D), *completeness* (C) and *informativeness* (I)—estimate this relationship by learning a *probe* $f$ to predict $z$ from $c$ and can be used to relate many other notions of disentanglement (see Locatello et al. 2020, § 6).

In this work, we extend this DCI framework in several ways. Our main idea is that *the functional capacity required to recover $z$ from $c$ is an important but thus-far neglected aspect of representation quality.* For example, consider the case of recovering $z$ from: (i) a noisy version thereof; (ii) raw, high-dimensional data (e.g. images); and (iii) a linearly-mixed version thereof, with each $c_i$ containing the same amount of information about each $z_j$ (precise definition in § 6.1). The noisy version (i) will do quite well with just linear capacity, but is fundamentally limited by the noise corruption; the raw data (ii) will likely do quite poorly with linear capacity, but eventually outperform (i) given sufficient capacity; and the linearly-mixed version (iii) will perfectly recover $z$ with just linear capacity, yet achieve the worst-possible disentanglement score of $D = 0$. Motivated by this observation, we introduce a measure of *explicitness* or *ease-of-use* based a representation's *loss-capacity curve* (see Fig. 1).

---

[*]Equal contribution.

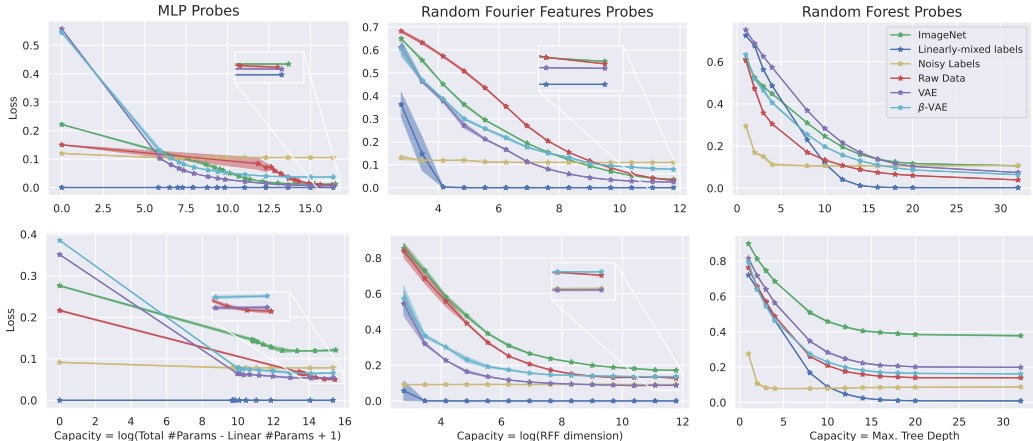

Figure 1: **Loss-capacity curves.** Empirical loss-capacity curves (see § 4.1) for various representations (see legend), datasets (top: MPI3D-Real, bottom: Cars3D), and probe types (left: multi-layer perceptrons / MLPs, middle: Random Fourier Features / RFFs, right: Random Forests / RFs). The loss was first averaged over factors $z_j$, and then means and 95% confidence intervals were computed over 3 random seeds. Details in § 6.

**Structure and contributions.** First, we connect the DCI metrics to two common notions of linear and nonlinear identifiability (§ 3). Next, we propose an extended DCI-ES framework (§ 4) in which we: (i) introduce two new complementary measures of representation quality—*explicitness* (E), derived from a representation's *loss-capacity curve*, and *size* (S); and then (ii) elucidate a means to compute the D and C scores for arbitrary black-box probes (e.g., MLPs). Finally, in our experiments (§ 6), we use our extended framework to compare different representations on the MPI3D-Real (Gondal et al., 2019) and Cars3D (Reed et al., 2015) datasets, illustrating the practical usefulness of our E score through its strong correlation with downstream performance.

## 2 BACKGROUND

Given a synthetic dataset of observations $x = g(z)$ along with the corresponding $K$-dimensional data-generating factors $z \in \mathbb{R}^K$, the DCI framework quantitatively evaluates an $L$-dimensional data representation or *code* $c = r(x) \in \mathbb{R}^L$ using two steps: (i) train a probe $f$ to predict $z$ from $c$, i.e., $\hat{z} = f(c) = f(r(x)) = f(r(g(z)))$; and then (ii) quantify $f$'s prediction error and its deviation from the ideal one-to-one mapping, namely a permutation matrix (with extra "dead" units in $c$ whenever $L > K$).[1] For step (i), Eastwood & Williams (2018) use Lasso (Tibshirani, 1996) or Random Forests (RFs, Breiman 2001) as linear or nonlinear predictors, respectively, for which it is straightforward to read-off suitable "relative feature importances".

**Definition 2.1.** $R \in \mathbb{R}^{L \times K}$ *is a* matrix of relative importances *for predicting $z$ from $c$ via $\hat{z} = f(c)$ if $R_{ij}$ captures some notion of the contribution of $c_i$ to predicting $z_j$ s.t. $\forall i, j$: $R_{ij} \geq 0$ and $\sum_{i=1}^{L} R_{ij} = 1$.*

For step (ii), Eastwood & Williams use $R$ and the prediction error to define and quantify three desiderata of disentangled representations: *disentanglement* (D), *completeness* (C), and *informativeness* (I).

**Disentanglement.** Disentanglement (D) measures the average number of data-generating factors $z_j$ that are captured by any single code $c_i$. The score $D_i$ is given by $D_i = 1 - H_K(P_{i.})$, where $H_K(P_{i.}) = -\sum_{k=1}^{K} P_{ik} \log_K P_{ik}$ denotes the entropy of the distribution $P_{i.}$ over *row* $i$ of $R$, with $P_{ij} = R_{ij} / \sum_{k=1}^{K} R_{ik}$. If $c_i$ is only important for predicting a single $z_j$, we get a perfect score of $D_i = 1$. If $c_i$ is equally important for predicting all $z_j$ (for $j = 1, \ldots, K$), we get the worst score of $D_i = 0$. The overall score D is then given by the weighted average $D = \sum_{i=1}^{L} \rho_i D_i$, with $\rho_i = \frac{1}{K} \sum_{k=1}^{K} R_{ik}$.

**Completeness.** Completeness (C) measures the average number of code variables $c_i$ required to capture any single $z_j$; it has also been called *compactness* (Ridgeway & Mozer, 2018). The score $C_j$ in capturing $z_j$ is given by $C_j = (1 - H_L(\tilde{P}_{.j}))$, where $H_L(\tilde{P}_{.j}) = -\sum_{\ell=1}^{L} \tilde{P}_{\ell j} \log_L \tilde{P}_{\ell j}$ denotes the

---

[1]W.l.o.g., it can be assumed that $z_i$ and $c_j$ are normalised to have mean zero and variance one for all $i, j$, for otherwise such normalisation can be "absorbed" into $g(\cdot)$ and $r(\cdot)$.

entropy of the distribution $\tilde{P}_j$ over *column* $j$ of $\boldsymbol{R}$, with $\tilde{P}_{ij} = R_{ij}$. If a single $c_i$ contributes to $z_j$'s prediction, we get a perfect score of $C_j = 1$. If all $c_i$ equally contribute to $z_j$'s prediction (for $i = 1, \ldots, L$), we get the worst score of $C_j = 0$. The overall completeness score is given by $C = \frac{1}{K} \sum_{j=1}^{K} C_j$.

**Remark 2.2.** *Together*, D and C quantify the degree of "mixing" between $\boldsymbol{c}$ and $\boldsymbol{z}$, i.e., the deviation from a one-to-one mapping. They are reported separately as they capture distinct criteria.

**Informativeness.** The informativeness (I) of representation $\boldsymbol{c}$ about data-generative factor $z_j$ is quantified by the prediction error, i.e., $I_j = 1 - \mathbb{E}[\ell(z_j, f_j(\boldsymbol{c}))]$, where $\ell$ is an appropriate loss function.[2] Note that $I_j$ depends on the capacity of $f_j$, as depicted in Fig. 1. Thus, for $I_j$ to accurately capture the informativeness of $\boldsymbol{c}$ about $z_j$, $f_j$ must have sufficient capacity to extract *all* of the information in $\boldsymbol{c}$ about $z_j$. This capacity-informativeness dependency motivates a separate measure of representation *explictness* in § 4.1. The overall informativeness score is given by $I = \frac{1}{K} \sum_{j=1}^{K} I_j$.

## 3 CONNECTION TO IDENTIFIABILITY

The goal of learning a data representation which recovers the underlying data-generating factors is closely related to blind source separation and independent component analysis (ICA, Comon 1994; Hyvärinen & Pajunen 1999; Hyvarinen et al. 2019). Whether a given learning algorithm provably achieves this goal up to acceptable ambiguities, subject to certain assumptions on the data-generating process, is typically formalised using the notion of *identifiability*. Two common types of identifiability for linear and nonlinear settings, respectively, are the following.

**Definition 3.1.** *We say that $\boldsymbol{c} = r(\boldsymbol{x}) = r(g(\boldsymbol{z}))$ identifies $\boldsymbol{z}$ up to sign and permutation if $\boldsymbol{c} = \boldsymbol{P}\boldsymbol{z}$ for some signed permutation matrix $\boldsymbol{P}$ (i.e., $|\boldsymbol{P}|$ is a permutation).*

**Definition 3.2.** *We say $\boldsymbol{c}$ identifies $\boldsymbol{z}$ up to permutation and element-wise reparametrisation if there exists a permutation $\pi$ of $\{1, ..., K\}$ and invertible scalar-functions $\{h_k\}_{k=1}^{K}$ s.t. $\forall j$: $c_j = h_j(z_{\pi(j)})$.*

We now establish theoretical connections between the DCI framework and these identifiability types.

**Proposition 3.3.** *If $D = C = 1$ and $K = L$ (i.e., $dim(\boldsymbol{c}) = dim(\boldsymbol{z})$), then $\boldsymbol{R}$ is a permutation matrix.*

All proofs are provided in Appendix A. Using Prop. 3.3, we can establish links to identifiability, provided the inferred representation $\boldsymbol{c}$ perfectly predicts the true data-generating factors $\boldsymbol{z}$, i.e., $I = 1$.

**Corollary 3.4.** *Under the same conditions as Prop. 3.3, if $\boldsymbol{z} = \boldsymbol{W}^\top \boldsymbol{c}$ (so that $I = 1$) for some $\boldsymbol{W}$ with $R_{ij} = \frac{|w_{ij}|}{\sum_{i=1}^{L} |w_{ij}|}$, then $\boldsymbol{c}$ identifies $\boldsymbol{z}$ up to permutation and sign (Defn. 3.1).*

For nonlinear $f$, we give a more general statement for suitably-chosen feature-importance matrices $\boldsymbol{R}$.

**Corollary 3.5.** *Under the same conditions as Prop. 3.3, let $\boldsymbol{z} = f(\boldsymbol{c})$ (so that $I = 1$) with $f$ an invertible and differentiable nonlinear function, and let $\boldsymbol{R}$ be a matrix of relative feature importances for $f$ (Defn. 2.1) with the property that $R_{ij} = 0$ if and only if $f_j$ does not depend on $c_i$, i.e., $\left\|\partial_i f_j\right\|_2 = 0$. Then $\boldsymbol{c}$ identifies $\boldsymbol{z}$ up to permutation and element-wise reparametrisation (Defn. 3.2).*

**Remark 3.6.** While the *if* part of Corollary 3.5 holds for most feature importance measures, the *only if* part, in general, does not: not using a feature $c_i$ is typically a *sufficient* condition for $R_{ij} = 0$, but it need not be a *necessary* condition (as required for Corollary 3.5). E.g., measures based on *average* performance may not satisfy this since a feature may not contribute on average, but still be used—sometimes helping and sometimes hurting performance (see § 7 for further discussion). In contrast, Gini importances, as used in random forests, *do* satisfy the necessary condition. While the non-invertibility of random forests prevents an explicit link to identifiability (typically studied for continuous features), they can still be a principled choice in practice (where features are often categorical).

**Summary.** We have established that the learnt representation $\boldsymbol{c}$ identifies the ground-truth $\boldsymbol{z}$ up to:

- sign and permutation if $D = C = I = 1$ and $f$ is linear;
- permutation and element-wise reparametrisation if $D = C = I = 1$ and $R_{ij} = 0 \Leftrightarrow \left\|\partial_i f_j\right\|_2 = 0$.

---

[2]Here we deviate from Eastwood & Williams (who had $I_j = \mathbb{E}[\ell(z_j, f_j(\boldsymbol{c}))]$) such that 1 is now the best score.

# 4 EXTENDED DCI-ES FRAMEWORK

Motivated by our theoretical insights from § 3—considering different probe function classes provides links to different types of identifiability—and the empirically-observed performance differences between representations trained with different-capacity probes shown in Fig. 1, we now propose several extensions of the DCI framework.

## 4.1 EXPLICITNESS (E)

We first introduce a new complementary notion of disentanglement based on the functional capacity required to recover or predict $z$ from $c$. The key idea is to measure the *explicitness* or *ease-of-use* (E) of a representation using its *loss-capacity curve*.

**Notation.** Let $\mathcal{F}$ be a probe function class (e.g., MLPs or RFs), let $f_j^* \in \arg\min_{f \in \mathcal{F}} \mathbb{E}[\ell(z_j, f(c))]$ be a minimum-loss probe for factor $z_j$ on a held-out data split[3], and let $\mathrm{Cap}(\cdot)$ be a suitable capacity measure on $\mathcal{F}$—e.g., for RFs, $\mathrm{Cap}(f)$ could correspond to the maximum tree-depth of $f$.

**Loss-capacity curves.** A loss-capacity curve for representation $c$, factor $z_j$, and probe class $\mathcal{F}$ displays test-set loss against probe capacity for increasing-capacity probes $f \in \mathcal{F}$ (see Fig. 1). To plot such a curve, we must train $T$ predictors with capacities $\kappa_1, \ldots, \kappa_T$ to predict $z_j$, with

$$f_j^t \in \arg\min_{f \in \mathcal{F}} \mathbb{E}\left[\ell(z_j, f(c))\right] \quad \text{s.t.} \quad \mathrm{Cap}(f) = \kappa_t. \tag{4.1}$$

Here $\kappa_1, \ldots, \kappa_T$ is a list of $T$ increasing probe *capacities*, ideally[4] shared by all representations, with suitable choices for $\kappa_1$ and $\kappa_T$ depending on both $\mathcal{F}$ and the dataset. For example, we may choose $\kappa_T$ to be large enough for all representations to achieve their lowest loss and, for random forest $f$s, we may choose an initial tree depth of $\kappa_1 = 1$ and then $T - 2$ tree depths between 1 and $\kappa_T$.

**AULCC.** We next define the *Area Under the Loss-Capacity Curve* (AULCC) for representation $c$, factor $z_j$, and probe class $\mathcal{F}$ as the (approximate) area between the corresponding loss-capacity curve and the loss-line of our best predictor $\ell_j^{*,c} = \mathbb{E}[\ell(z_j, f_j^*(c))]$. To compute this area, depicted in Fig. 2, we use the trapezoidal rule

$$\mathrm{AULCC}(z_j, c; \mathcal{F}) = \sum_{t=2}^{t^{*,c}} \left( \frac{1}{2} \left( \ell_j^{t-1,c} + \ell_j^{t,c} \right) - \ell_j^{*,c} \right) \cdot \Delta \kappa_t,$$

where $t^{*,c}$ denotes the index of $c$'s lowest-loss capacity $\kappa_{*,c}$; $\ell_j^{t,c} = \mathbb{E}[\ell(z_j, f_j^t(c))]$ the test-set loss with predictor $f_j^t$, see Eq. (4.1); and $\Delta \kappa_t = \kappa_t - \kappa_{t-1}$ the size of the capacity interval at step $t$. If the lowest loss is achieved at the lowest capacity, i.e. $t^{*,c} = 1$, we set $\mathrm{AULCC} = 0$.

**Explicitness.** We define the **explicitness** (E) of representation $c$ for predicting factor $z_j$ with predictor class $\mathcal{F}$ as

$$E(z_j, c; \mathcal{F}) = 1 - \frac{\mathrm{AULCC}(z_j, c; \mathcal{F})}{\frac{1}{2}(\kappa_T - \kappa_1)(\ell_j^b - \ell_j^*)},$$

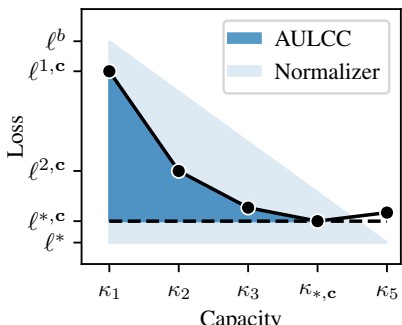

Figure 2: **Explicitness via the area under the loss-capacity curve (AULCC).** Here, $\kappa_1, \ldots, \kappa_T$ (x-axis) are a sequence of increasing function-capacities and $\ell^{1,c}, \ldots, \ell^{T,c}$ (y-axis) are the losses achieved by the corresponding optimal predictors for $c$. The lowest loss $\ell^{*,c}$ is achieved at capacity $\kappa_{*,c}$, while $\ell^b$ and $\ell^*$ are suitable baseline and best-possible losses for the probe class.

where $\ell_j^b$ is a suitable baseline loss (e.g., that of $\mathbb{E}[z_j]$) and $\ell_j^*$ a suitable lowest loss (e.g., 0) for $\mathcal{F}$. Here, the denominator represents the area of the light-blue triangle in Fig. 2, *normalizing* the AULCC such that $E_j \in [-1, 1]$ so long as $\ell_j^* < \ell_j^b$. The best score $E_j = 1$ means that the best loss was achieved with the lowest-capacity probe $f_j^1$, i.e., $\ell_j^{*,c} = \ell_j^{1,c}$ and $\kappa_{*,c} = \kappa_1$, and thus our representation $c$ was explicit or easy-to-use for predicting $z_j$ with $f \in \mathcal{F}$ since there was *no surplus*

---

[3]In practice, all expectations are taken w.r.t. the corresponding empirical (train/validation/test) distributions.
[4]True for RFs but not input-size dependent MLPs (see § 6).

*capacity required (beyond $\kappa_1$) to achieve our lowest loss*. In contrast, $E_j = 0$ means that the loss reduced *linearly* from $\ell_j^b$ to $\ell_j^*$ with increased probe capacity, i.e., AULCC = Normalizer in Fig. 2. More generally, if $\ell^{*,c} = \ell^*$, i.e. the lowest loss for $\mathcal{F}$ can be reached with representation $c$, then $E_j < 0$ implies that the loss decreased *sub-linearly* with increased capacity while $E_j > 0$ implies it decreased *super-linearly*. The overall explicitness score is given by $E = \frac{1}{K}\sum_{j=1}^{K} E_j$.

**E vs. I.** While the informativeness score $I_j$ captures the (total) amount of information in $c$ about $z_j$, the explicitness score $E_j$ captures the *ease-of-use* of this information. In particular, while $I_j$ is quantified by the *lowest prediction error with any capacity* $\ell^{*,c}$, corresponding to a single point on $c$'s loss-capacity curve, $E_j$ is quantified by the *area under this curve*.

**A fine-grained picture of identifiability.** Compared to the commonly-used mean correlation coefficient (MCC) or Amari distance (Amari et al., 1996; Yang & Amari, 1997), the $D, C, I, E$ scores represent empirical measures which: (i) easily extend to mismatches in dimensionalities, i.e., $L > K$; and (ii) provide a more fine-grained picture of identifiability (violations), for if the initial probe capacity $\kappa_1$ is linear and $R$ satisfies Corollary 3.5, we have that:

- $D\!=\!C\!=\!I\!=\!E\!=\!1 \implies$ identified up to sign and permutation (Defn. 3.1);
- $D\!=\!C\!=\!I\!=\!1 \implies$ identified up to permutation and element-wise reparametrisation (Defn. 3.2);
- $I\!=\!E\!=\!1 \implies$ identified up to invertible linear transformation (cf. Khemakhem et al., 2020).

Thus, if $D\!=\!C\!=\!I\!=\!E\!=\!1$ does not hold exactly, which score deviates the most from 1 may provide valuable insight into the type of identifiability violation.

**Probe classes.** As emphasized above, whether or not a representation $c$ is explicit or easy-to-use for predicting factor $z_j$ depends on the class of probe $\mathcal{F}$ used, e.g., MLPs or RFs. More generally, the explicitness of a representation depends on the way in which it is used in downstream applications, with different downstream uses or probe classes resulting in different definitions of explicit or easy-to-use information. We thus conduct experiments with different probe classes in § 6.

## 4.2    Size (S)

We next introduce a measure of representation size (S), motivated by the observation that larger representations tend to be both more informative and more explicit (see Tab. 1, more details below). Reporting S thus allows size-informativeness and size-explicitness trade-offs to be analysed.

**A measure of size.** We measure representation *size* (S) relative to the ground-truth as:

$$S = \frac{K}{L} = \frac{\dim(z)}{\dim(c)}.$$

When $L \geq K$, as often the case, we have $S \in (0, 1]$ with the perfect score being $S\!=\!1$. However, if we also consider the $L < K$ case, which would likely sacrifice some informativeness, we have $S \in (1, K]$.

**Larger representations are often more *informative*.** When $L < K$, it is intuitive that larger representations are more informative—they can simply preserve more information about $z$. When $L > K$, however, it is also common for larger representations to be more informative, perhaps due to an easier optimization landscape (Frankle & Carbin, 2019; Golubeva et al., 2021). Tab. 1 illustrates this point, where AE-5 denotes an autoencoder with $L\!=\!5$. Note that $K\!=\!7$ for MPI3D-Real (see § 6).

**Larger representations are often more *explicit*.** The explicitness of a representation also depends on its size: larger representations tend to be more explicit, as is apparent from the second column of Tab. 1. To explain this, we plot the corresponding loss-capacity curves in Fig. 3. Here we see that the increased explicitness (i.e., smaller AULLC) of larger representations stems from a substantially lower initial loss when using a linear-capacity MLP probe. The fact that larger representations perform better with linear-capacity MLPs is unsurprising since they have more parameters.

## 4.3    Probe-agnostic feature importances

Finally, to meaningfully discuss more flexible probe-function choices within the DCI-ES framework, we point out that the D and C scores can be computed for arbitrary black-box probes $f$ by using *probe-agnostic* feature-importance measures. In particular, in our experiments (§ 6), we use SAGE (Covert et al., 2020) which summarises each feature's importance based on its contribution to

| Representation | I | E | S |
|:---:|:---:|:---:|:---:|
| AE-5 | 0.75 | 0.74 | 1.4 |
| AE-7 | 0.92 | 0.71 | 1.0 |
| AE-10 | 0.99 | 0.72 | 0.7 |
| AE-100 | 1.0 | 0.90 | 0.07 |
| AE-500 | 1.0 | 0.93 | 0.01 |

Table 1: I, E and S scores for auto-encoders of various sizes on `MPI3D-Real` with MLP probes.

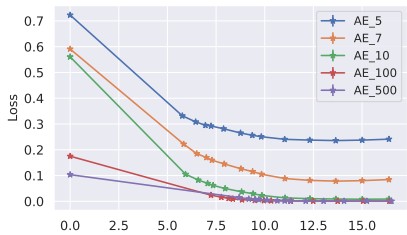

Figure 3: Loss-capacity curves for auto-encoders of various sizes on `MPI3D-Real` with MLP probes.

predictive performance, making use of Shapley values (Shapley, 1953) to account for complex feature interactions. Such probe-agnostic measures allow the $D$ and $C$ scores to be computed for probes with no inherent or built-in notion of feature importance (e.g., MLPs), thereby generalising the Lasso and RF examples of Eastwood & Williams (2018, § 4.3). While SAGE has several practical advantages over other probe-agnostic methods (see, e.g., Covert et al., 2020, Table 1), it may not satisfy the conditions required to link the $D$ and $C$ scores to different identifiability equivalence classes (see Remark 3.6). Future work may explore alternative methods which do, e.g., by looking at a feature's mean *absolute* attribution value (Lundberg & Lee, 2017) since, intuitively, absolute contributions do not allow for a cancellation of positive and negative attribution on average (cf. Remark 3.6).

## 5 RELATED WORK

**Explicit representations.** Eastwood & Williams (2018, § 2) noted that the informativeness score with a linear probe quantifies the amount of information in $c$ about $z$ that is "explicitly represented", while Ridgeway & Mozer (2018, § 3) proposed a measure of "explicitness" which simply reports the informativeness score with a linear probe. In contrast, our DCI-ES framework differentiates between the amount of information in $c$ about $z$ (*informativeness*) and the ease-of-use of this information (*explicitness*). This allows a more fine-grained analysis of the relationship between $c$ and $z$, both theoretically (distinguishing between more identifiability equivalence classes; § 3) and empirically (§ 6).

**Loss-capacity curves.** Plotting loss against model complexity or capacity has long been used in statistical learning theory, e.g., for studying the bias-variance trade-off (Hastie et al., 2009, Fig. 7.1). More recently, such loss-capacity curves have been used to study the double-descent phenomenon of neural networks (Belkin et al., 2019; Nakkiran et al., 2021) as well as the scaling laws of large language models (Kaplan et al., 2020). However, they have yet to be used for assessing the quality or explicitness of representations.

**Loss-data curves.** Whitney et al. (2020) use loss-data curves, which plot loss against dataset size, to assess representations. They measure the quality of a representation by the *sample complexity* of learning probes that achieve low loss on a task of interest. Loss-data curves are also studied under the term *learning curves* in standard/purely supervised-learning settings (see, e.g., Viering & Loog, 2021, for a recent review). In contrast, we focus on *functional complexity* and the task of predicting the data-generative factors $z$, and then discuss the functional complexity for other tasks $y$ in § 7.

## 6 EXPERIMENTS

### 6.1 SETUP

**Data.** We perform our analysis of loss-capacity curves on the `MPI3D-Real` (Gondal et al., 2019) and `Cars3D` (Reed et al., 2015) datasets. `MPI3D-Real` contains $\approx 1M$ real-world images of a robotic arm holding different objects with seven annotated ground-truth factors: object colour (6), object shape (6), object size (2), camera height (3), background colour (3) and two degrees of rotations of the arm ($40 \times 40$); numbers in brackets indicate the number of possible values for each factor. `Cars3D` contains $\approx 17.5k$ rendered images of cars with three annotated ground-truth factors: camera elevation (4), azimuth (24) and car type (183).

**Representations.** We use the following synthetic baselines and standard models as representations:

- *Noisy labels:* $c = z + \epsilon$, with $\epsilon \sim \mathcal{N}(\mathbf{0}, 0.01 \cdot \mathbf{I}_K)$.
- *Linearly-mixed labels:* $c = Wz$, with $W_{ij} = \frac{1}{LK} + \epsilon_{ij}$ and $\epsilon_{ij} \sim \mathcal{N}(0, 0.001)$ to achieve "uniform mixing" (each $z_j$ evenly-distributed across the $c_i$s) while also ensuring the invertibility of $W$ a.s.
- *Raw data (pixels):* $c = x = g(z)$.
- *Others:* We also use VAEs (Kingma & Welling, 2014) with 10 latents ($L$=10), $\beta$-VAEs (Higgins et al. 2017, $L$=10); and an ImageNet-pretrained ResNet18 (He et al. 2016, $L$=512).

**Probes.** We use MLPs, RFs and Random Fourier Features (RFFs, Rahimi & Recht 2007) to predict $z$ from $c$, with RFFs having a linear classifier on top. For MLPs, we start with linear probes (no hidden layers) then increase capacity by adding two hidden layers and varying their widths from $2 \times K$ to $512 \times K$. We then measure capacity based on the number of "extra" parameters beyond that of the linear probe, and compute feature importances using SAGE with permutation-sampling estimators and marginal sampling of masked values (see https://github.com/iancovert/sage). For RFs, we use ensembles of 100 trees, control capacity by varying the maximum depth between 1 and 32, and compute feature importances using Gini importance. For RFFs, we control capacity by exponentially increasing the number of random features from $2^4$ to $2^{17}$, and compute feature importances using SAGE.

**Implementation details.** We split the data into training, validation and test sets of size 295k, 16k, and 726k respectively for `MPI3D-Real` and 12.6k, 1.4k, 3.4k for `Cars3d`. We use the validation split for hyperparameter selection and report results on the test split. We train MLP probes using the Adam (Kingma & Ba, 2015) optimizer for 100 epochs. We use mean-square error and cross-entropy losses for continuous and discrete factors $z_j$, respectively. To compute $E_j$, we use the baseline losses of $\mathbb{E}[z_j]$ and a random classifier for continuous and discrete $z_j$, respectively. Further details can be found in our open-source code: https://github.com/andreinicolicioiu/DCI-ES.

## 6.2 EVALUATION RESULTS: CURVES AND SCORES

**Loss-capacity curves.** Fig. 1 depicts loss-capacity curves for the three probes and two datasets, averaged over factors $z_j$. In all six plots, the noisy-labels baseline performs well with low-capacity and then is surpassed by other representations given sufficient capacity, as expected. Note that the linearly-mixed-labels baseline immediately achieves $\approx 0$ loss with MLP probes but not with RFF or RF probes, supporting the idea that the explicitness or ease-of-use of a representation depends on the way in which it is used. Also note that, with MLP probes and $\log(\text{excess \#params})$ as the capacity measure, larger input representations are afforded more parameters with a linear probe and thus are more expressive. This further explains why larger representations are often more explicit, and highlights the difficulty of measuring the capacity of MLPs—an active area of research in its own right, which we discuss in § 7. Finally, in Appendix B.2, we investigate the effect of dataset size by plotting loss-capacity curves for different dataset sizes, observing that larger datasets have smaller performance gaps between: (i) synthetic and learned representations; and (ii) small and large representations (see Fig. 10).

**DCI-ES scores.** Tab. 2 reports the corresponding DCI-ES scores, along with some oracle scores for MLPs. Note that: (i) the GT labels $z$ get perfect scores of 1 for all metrics; (ii) by attaining very low D and C scores but near-perfect E scores, the linearly-mixed labels expose the key difference between mixing-based (D,C) and functional-capacity-based (E) measures of the *simplicity of the c-z relationship*; (iii) larger representations (ImgNet-pretr, raw data) tend to be more explicit than smaller ones (VAE, $\beta$-VAE), with S and E together capturing this size-explicitness trade-off; and (iv) $\beta$-VAE achieves better mixing-based scores (D,C) but similar E scores compared to the VAE, illustrating that these two "disentanglement" notions are indeed orthogonal and complementary.

## 6.3 DOWNSTREAM RESULTS: SCORE CORRELATIONS

**Setup.** To illustrate the practical usefulness of our explictness score, we calculate its correlation with downstream performance when using low-capacity probes. Using `MPI-3D`, we create 14 synthetic downstream tasks: 7 regression tasks with $y^i = M^i \mathbf{z}$ and $M^i_{jk} \sim U(0,1)$, and 7 classification tasks with $y^i = \mathbb{1}_{\{z_i > m_i\}}$ and $m_i$ the median value of factor $z_i$. For representations, we use AEs, VAEs and $\beta$-VAEs, 2 latent dimensionalities (i.e. $\dim(\mathbf{c})$) of 10 and 50, and 5 random seeds—resulting in a total of 30 different representations $\mathbf{c}$. To compute the correlations, we first compute the DCIE scores as before, training MLP and RF probes $f$ to predict $\mathbf{z}$ from $\mathbf{c}$, i.e. $\hat{z}_j = f_j(\mathbf{c})$, and then compute the down-

Table 2: **DCI-ES scores for different probes, datasets and representations.** Empirical scores using MLP, RFF and RF probes trained on the `MPI3D-Real` and `Cars3D` datasets, as well as theoretical/oracle scores for some simple representations with MLPs (MLP*). We show averages over 3 random seeds; standard deviations were all $< 0.05$. Note that which representation is deemed "best" depends on the application of interest—some are more disentangled, some more informative, some more explicit, etc.

| Representation | Probe | MPI3D | | | | | CARS3D | | | | |
|---|---|---|---|---|---|---|---|---|---|---|---|
| | | **D** | **C** | **I** | **E** | **S** | **D** | **C** | **I** | **E** | **S** |
| GT Labels $z$ | MLP* | 1 | 1 | 1 | 1 | 1 | 1 | 1 | 1 | 1 | 1 |
| Noisy labels | MLP* | 1 | 1 | 0.9 | 1 | 1.0 | 1 | 1 | 0.9 | 1 | 1.0 |
| | MLP | 0.97 | 0.97 | 0.89 | 0.99 | 1.0 | 0.99 | 0.99 | 0.92 | 0.99 | 1.0 |
| | RFF | 0.97 | 0.97 | 0.88 | 0.99 | 1.0 | 1.0 | 1.0 | 0.91 | 1.0 | 1.0 |
| | RF | 0.93 | 0.93 | 0.89 | 0.98 | 1.0 | 0.95 | 0.95 | 0.92 | 0.99 | 1.0 |
| Linearly-mixed labels | MLP* | 0 | 0 | 1 | 1 | 1.0 | 0 | 0 | 1 | 1 | 1.0 |
| | MLP | 0.13 | 0.22 | 1.0 | 1.0 | 1.0 | 0.21 | 0.22 | 1.0 | 1.0 | 1.0 |
| | RFF | 0.11 | 0.21 | 1.0 | 0.94 | 1.0 | 0.19 | 0.19 | 1.0 | 1.0 | 1.0 |
| | RF | 0.17 | 0.21 | 1.0 | 0.72 | 1.0 | 0.08 | 0.12 | 0.99 | 0.78 | 1.0 |
| VAE | MLP | 0.15 | 0.14 | 0.99 | 0.71 | 0.7 | 0.18 | 0.11 | 0.95 | 0.80 | 0.3 |
| | RFF | 0.13 | 0.14 | 0.97 | 0.69 | 0.7 | 0.16 | 0.11 | 0.91 | 0.87 | 0.3 |
| | RF | 0.10 | 0.10 | 0.93 | 0.65 | 0.7 | 0.14 | 0.09 | 0.80 | 0.81 | 0.3 |
| $\beta$-VAE | MLP | 0.46 | 0.41 | 0.96 | 0.74 | 0.7 | 0.27 | 0.23 | 0.94 | 0.78 | 0.3 |
| | RFF | 0.41 | 0.38 | 0.92 | 0.71 | 0.7 | 0.31 | 0.23 | 0.86 | 0.86 | 0.3 |
| | RF | 0.39 | 0.35 | 0.94 | 0.76 | 0.7 | 0.20 | 0.17 | 0.84 | 0.83 | 0.3 |
| ImgNet-pretr | MLP | 0.16 | 0.10 | 0.99 | 0.82 | 0.01 | 0.22 | 0.07 | 0.88 | 0.86 | 0.006 |
| | RFF | 0.15 | 0.13 | 0.96 | 0.58 | 0.01 | 0.24 | 0.10 | 0.83 | 0.65 | 0.006 |
| | RF | 0.35 | 0.20 | 0.89 | 0.78 | 0.01 | 0.20 | 0.09 | 0.62 | 0.83 | 0.006 |
| Raw data | MLP | 0.22 | 0.16 | 0.99 | 0.82 | 0.001 | 0.39 | 0.27 | 0.95 | 0.84 | 0.0002 |
| | RFF | 0.37 | 0.14 | 0.97 | 0.44 | 0.001 | 0.32 | 0.24 | 0.87 | 0.64 | 0.0002 |
| | RF | 0.84 | 0.41 | 0.96 | 0.80 | 0.001 | 0.53 | 0.31 | 0.86 | 0.82 | 0.0002 |

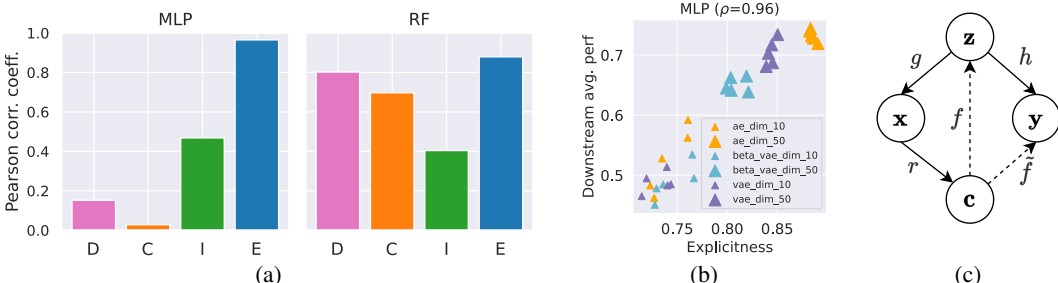

Figure 4: (a) Correlation coefficients $\rho$ between DCIE scores and downstream performance with low-capacity probes. (b) E vs. downstream performance with linear MLPs. (c) DCIE scores are computed by predicting **z** from **c** with probes $f$, then downstream tasks $y = h(\mathbf{z})$ are solved by predicting $y$ from **c** with low-capacity probes $\tilde{f}$.

stream performance by training new low-capacity MLP and RF probes $\tilde{f}$ to predict $y$ from **c**, i.e. $\hat{y}^i = \tilde{f}_i(\mathbf{c})$ (see Fig. 4c). For MLP probes, low capacity means linear. For RF probes, low capacity means the maximum tree depth is 10. Next, we average the downstream performances across all 14 tasks before computing the correlation coefficient between this average and each of the D, C, I, and E scores.

**Analysis.** Figs. 4a and 4b show that E is strongly correlated with downstream performance when using both MLP ($\rho = 0.96$, $p = 8\text{e-}18$) and RF probes ($\rho = 0.88$, $p = 2\text{e-}10$). In contrast, mixing-based disentanglement scores (D, C) exhibit much weaker correlations with MLP probes, corroborating the results of Träuble et al. (2022, Fig. 8) who also found a weak correlation between D and downstream performance on reinforcement learning tasks with MLPs. See App. B.1 for further details and results.

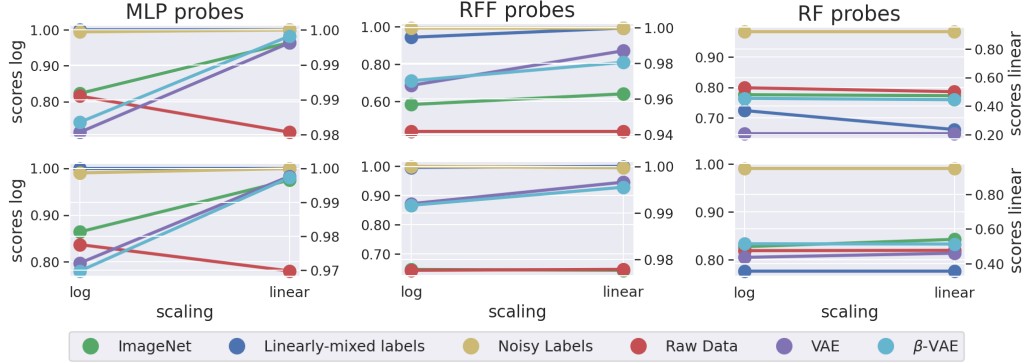

Figure 5: Explicitness scores on `MPI3D-Real` (top) and `Cars3D` (bottom) for different representations (see legend). Each pair of points represents the score with logarithmic (left) and linear (right) capacity-scaling.

## 7 DISCUSSION

**Why connect disentanglement and identifiability?** Connecting prediction-based evaluation in the disentanglement literature to the more theoretical notion of identifiability has several benefits. Firstly, it provides a concrete link between two often-separate communities. Secondly, it endows the often empirically-driven or practice-focused disentanglement metrics with a solid and well-studied theoretical foundation. Thirdly, compared to the commonly-used MCC or Amari distance, it provides the ICA or identifiability community with more fine-grained empirical measures, as discussed in § 4.1.

**Measuring probe capacity.** Our measure of explicitness $E$ depends strongly on the choice of capacity measure for a probe or function class. For some probes like RFs or RFFs, there exist natural measures of capacity. However, for other probes like MLPs, coming up with a good capacity measure is itself an important and active area of research (Jiang et al., 2020; Dziugaite et al., 2020). Another difficulty arises from choosing a capacity *scale*, with different scales (e.g., log, linear, etc.) leading to loss-capacity curves with different shapes, areas and thus explicitness scores. To investigate the extent of this issue, i.e., the sensitivity of our explicitness measure to the choice of capacity scale, Fig. 5 compares the explicitness scores when using logarithmic and linear scaling. Here we see that the *ranking* essentially remains the same except for the raw-data representation with MLP probes.

**Measuring feature importance.** Similarly, the choice of feature-importance measure has a strong influence on the $D$ and $C$ scores, with some probes having natural or in-built measures (e.g., random forests) and others not (e.g., MLPs). For the latter, we proposed the use of probe-agnostic feature-importance measures like SAGE, and specified the conditions (Corollary 3.5) that importance measures must satisfy if the resulting $D$ and $C$ scores are to be connected to identifiability. As with probe capacity, coming up with good measures of feature importance is its own orthogonal field of study (e.g., model explainability), with future advances likely to improve the DCI-ES framework.

**What about explicitness for other tasks $y$?** While we focused on the explicitness or ease-of-use of a representation for predicting the data-generative factors $z$, one may also be interested in its ease-of-use for other tasks/labels $y$. While it is often implicitly assumed that the ease-of-use for predicting $z$ correlates with the ease-of-use for common tasks of interest (e.g., object classification, segmentation, etc.), future work could directly evaluate the explicitness of a representation for particular tasks $y$. For example, one could consider the entire loss-capacity curve when benchmarking self-supervised representations on ImageNet, rather than just linear-probe performance (a single slice). Future work could also explore the trade-off between *explicit but task-specific* and *implicit but task-agnostic* representations.

## 8 CONCLUSION

We have presented DCI-ES—an extended disentanglement framework with two new complementary measures of representation quality—and proven its connections to identifiability. In particular, we have advocated for additionally measuring the explicitness (E) of a representation by the functional capacity required to use it, and proposed to quantify this explicitness using a representation's loss-capacity curve. Together with the size (S) of a representation, we believe that our extended DCI-ES framework allows for a more fine-grained and nuanced benchmarking of representation quality.

ACKNOWLEDGMENTS

The authors would like to thank Chris Williams, Francesco Locatello, Nasim Rahaman, Sidak Singh and Yash Sharma for helpful discussions and comments. This work was supported by the German Federal Ministry of Education and Research (BMBF): Tübingen AI Center, FKZ: 01IS18039A, 01IS18039B; and by the Machine Learning Cluster of Excellence, EXC number 2064/1 – Project number 390727645.

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

# A   PROOFS

## A.1   PROOF OF PROPOSITION 3.3

**Proposition 3.3.** *If $D = C = 1$ and $K = L$ (i.e., $dim(\mathbf{c}) = dim(\mathbf{z})$), then $\mathbf{R}$ is a permutation matrix.*

*Proof.* First, by Defn. 2.1, we have $0 \leq R_{ij}$ and $\sum_{i=1}^{L} R_{ij} = 1 \; \forall i, j$, so $0 \leq R_{ij} \leq 1$. It follows that $\forall i, j : P_{i\cdot}, \tilde{P}_{\cdot j} \in \Delta_{K-1}$, where $\Delta_{K-1}$ denotes the $K$-dim. probability simplex, i.e., $P_{i\cdot}$ and $\tilde{P}_{\cdot j}$ are valid probability vectors. Hence, the Shannon entropies $H_K(P_{i\cdot}), H_K(\tilde{P}_{\cdot j})$ are well-defined $\forall i, j$, and, due to using $\log_K$ in the definition of $H_K$ (see § 2), are bounded in $[0, 1]$. It follows that $\forall i, j : 0 \leq D_i \leq 1$ and $0 \leq C_j \leq 1$. Since $D$ and $C$ are convex combinations of the $D_i$ and $C_j$, we have

$$D = 1 \iff \forall i : D_i = 1 \iff \forall i : H_K(P_{i\cdot}) = 0 \,,$$
$$C = 1 \iff \forall j : C_j = 1 \iff \forall j : H_K(\tilde{P}_{\cdot j}) = 0 \,.$$

Now for any $\mathbf{p} = (p_1, ..., p_K) \in \Delta_{K-1}$, we have that

$$H_K(\mathbf{p}) = -\sum_{k=1}^{K} p_k \log_K p_k = 0 \iff \forall k : p_k \log_K p_k = 0 \iff \forall k : p_k \in \{0,1\}$$

where $p_k \log p_k := 0$ for $p_k = 0$, consistent with $\lim_{x \to 0^+} x \log x = 0$. Together with the simplex constraint, this implies that $\mathbf{p}$ must be a standard basis vector $\mathbf{p} = \mathbf{e}_l$ for some $l$, i.e., $p_l = 1$ and $p_k = 0$ for $k \neq l$. Hence, $P_{i\cdot}, \tilde{P}_{\cdot j}$ must be standard basis vectors for all $i, j$, and so each row and column of $\mathbf{R}$ contains exactly one non-zero element. Since columns of $\mathbf{R}$ sum to one, these non-zero elements must all be one. $\qquad\square$

## A.2   PROOF OF COROLLARY 3.4

**Corollary 3.4.** *Under the same conditions as Prop. 3.3, if $\mathbf{z} = \mathbf{W}^\top \mathbf{c}$ (so that $I = 1$) for some $\mathbf{W}$ with $R_{ij} = \dfrac{|w_{ij}|}{\sum_{i=1}^{L} |w_{ij}|}$, then $\mathbf{c}$ identifies $\mathbf{z}$ up to permutation and sign (Defn. 3.1).*

*Proof.* First, we show that $R_{ij} = \dfrac{|w_{ij}|}{\sum_{i=1}^{L} |w_{ij}|}$ is a well-defined feature importance matrix. Suppose for a contradiction, that $\sum_{i=1}^{L} |w_{il}| = 0$ for some $l$. Since $|w_{il}| \geq 0$, this implies $w_{il} = 0$ for all $i$. Consider $z_l = \sum_{i=1}^{L} w_{il} c_i$. Taking the covariance, we obtain $\text{Var}[z_l] = \sum_{i,j=1}^{L} w_{il} w_{jl} \text{Cov}(c_i, c_j) = 0$, which is a contradiction since $z_l$ has positive (unit) variance by the normalisation assumption (see footnote 1). Hence, $\sum_{i=1}^{L} |w_{il}| > 0$ for all $l$. Thus $\mathbf{R}$ is well-defined, with its elements being non-negative and its columns summing to one by construction, so it is a valid feature importance matrix.

Next, note that we can write $\mathbf{R} = |\mathbf{W}|\mathbf{D}$ where $\mathbf{D}$ is the invertible diagonal matrix with positive diagonal entries $D_{jj} = \dfrac{1}{\sum_{i=1}^{L} |w_{ij}|} > 0$.

By Prop. 3.3, $\mathbf{R}$ is a permutation matrix, so $\mathbf{R} = \mathbf{P} = |\mathbf{W}|\mathbf{D}$ for some permutation matrix $\mathbf{P}$. Right multiplication by $\mathbf{D}^{-1}$ yields $\mathbf{P}\mathbf{D}^{-1} = |\mathbf{W}|$, that is $|\mathbf{W}|$ has exactly one non-zero, positive element in each row and each column (and zeros elsewhere). Thus $\mathbf{W}$ and therefore also $\mathbf{W}^\top$ are generalised permutation matrices. Hence $(\mathbf{W}^\top)^{-1}$ exists and is also a generalised permutation matrix.

Finally, consider $\mathbf{c} = (\mathbf{W}^\top)^{-1}\mathbf{z}$. Since all but ony element in each row of $(\mathbf{W}^\top)^{-1}$ are zero, we have for any $i$: $c_i = \tilde{w}_{ij} z_j$ for some $j$, where $\tilde{w}_{ij}$ denotes the $(i, j)$ element of $(\mathbf{W}^\top)^{-1}$. By considering the variances of both sides and recalling that all $c_i$'s and $z_j$'s are normalised to unit variance, it follows that $1 = \text{Var}(c_i) = \tilde{w}_{ij}^2 \text{Var}(z_j) = \tilde{w}_{ij}^2$. Hence, $\tilde{w}_{ij}^2 = \pm 1$ and so $(\mathbf{W}^\top)^{-1}$ is, in fact, a signed permutation matrix, which concludes the proof. $\qquad\square$

## A.3   PROOF OF COROLLARY 3.5

**Corollary 3.5.** *Under the same conditions as Prop. 3.3, let $\mathbf{z} = f(\mathbf{c})$ (so that $I = 1$) with $f$ an invertible and differentiable nonlinear function, and let $\mathbf{R}$ be a matrix of relative feature importances*

for $f$ (Defn. 2.1) with the property that $R_{ij} = 0$ if and only if $f_j$ does not depend on $c_i$, i.e., $\left\lVert \partial_i f_j \right\rVert_2 = 0$. Then $\boldsymbol{c}$ identifies $\boldsymbol{z}$ up to permutation and element-wise reparametrisation (Defn. 3.2).

*Proof.* For any $j$ consider $z_j = f_j(\boldsymbol{c})$. By Prop. 3.3, $R$ is a permutation matrix, so column $j$ of $\boldsymbol{R}$ contains exactly one non-zero entry in row $\pi(j)$ for some permutation $\pi$ of $\{1, ..., K\}$. Hence, by the assumed property of $R$, $f_j(\boldsymbol{c})$ does not depend on $c_i$ for all $i \neq \pi(j)$, and thus $z_j = f_j(c_{\pi(j)}) \; \forall j$. By invertibility of $f$, we obtain $c_j = h_j(z_{j'})$ with $h_j = f_{j'}^{-1}$ and $j' = \pi^{-1}(j)$. $\qquad\square$

## B  ADDITIONAL EXPERIMENTAL RESULTS

### B.1  DOWNSTREAM CORRELATIONS

Here we present the full results of the correlations between the DCIE scores and downstream performance, the latter with low-capacity probes (as discussed in § 6.3).

In Tab. 3 and Tab. 4 we show the values of the Pearson and Spearman correlations alongside the corresponding $p$-values[5]. Note that some of the assumptions behind these $p$-values, e.g. that the DCIE scores and downstream performances are normally distributed, likely do not hold. Thus, these $p$-values should not be interpreted as precise probabilities but rather as rough indications of statistical significance. In Tab. 5 we show the correlations for regression and classification tasks separately, with both task types exhibiting similar correlations. We note that E has the strongest correlation with the downstream performance (when using low-capacity probes for the downstream task).

Table 3: Pearson correlation coefficient $\rho$ between the D, C, I, and E scores and downstream performance, along with the corresponding $p$-values (in parentheses). See § 6.3 for experimental details.

| Probe $f$ | D | C | I | E |
|---|---|---|---|---|
| MLP | 0.15 ($4 \times 10^{-1}$) | 0.28 ($9 \times 10^{-1}$) | 0.47 ($9 \times 10^{-3}$) | **0.96** ($8 \times 10^{-18}$) |
| RF | 0.8 ($1 \times 10^{-7}$) | 0.70 ($2 \times 10^{-5}$) | 0.4 ($3 \times 10^{-2}$) | **0.88** ($2 \times 10^{-10}$) |

Table 4: Spearman rank correlation between the D, C, I, and E scores and downstream performance, along with the corresponding $p$-values (in parentheses).

| Probe $f$ | D | C | I | E |
|---|---|---|---|---|
| MLP | 0.12 ($5 \times 10^{-1}$) | -0.07 ($7 \times 10^{-1}$) | 0.55 ($2 \times 10^{-3}$) | **0.94** ($1 \times 10^{-14}$) |
| RF | **0.81** ($6 \times 10^{-8}$) | 0.75 ($2 \times 10^{-6}$) | 0.28 ($1 \times 10^{-1}$) | 0.78 ($3 \times 10^{-7}$) |

Table 5: Pearson correlation coefficient $\rho$ between the D,C,I, E scores and downstream performance for each task type (regression and classification). Correlations are similar across both task types.

| Probe $f$ | Task | D | C | I | E |
|---|---|---|---|---|---|
| MLP | Regression | 0.16 | 0.04 | 0.46 | **0.96** |
| | Classification | 0.14 | 0.00 | 0.48 | **0.96** |
| RF | Regression | 0.76 | 0.66 | 0.42 | **0.84** |
| | Classification | 0.81 | 0.72 | 0.35 | **0.89** |

**Score-by-score analysis.** To get a deeper insight into the correlations reported in Tabs. 3 and 4, we plot each of the D, C, I and E scores against downstream performance for each of the 30 models considered in § 6.3. As shown in Figs. 6 to 9, only E correlates strongly with downstream performance

---

[5]Computed using `https://docs.scipy.org/doc/scipy/reference/generated/scipy.stats.pearsonr.html`

for both probe types, again highlighting: (i) the value that E adds to the existing DCI framework; and (ii) the practical usefulness of reporting E when comparing/evaluating learned representations.

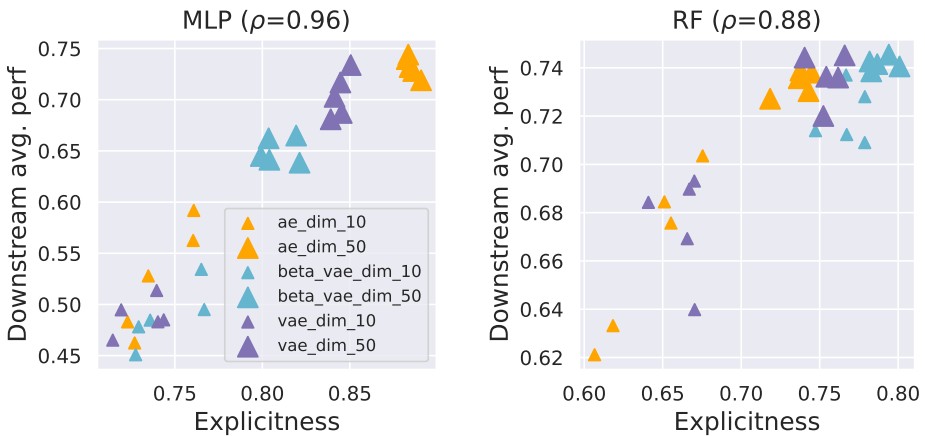

Figure 6: **Explicitness (E) vs. downstream performance.** Scatter plots show 30 data points: 3 models (AEs, VAEs, $\beta$-VAEs) $\times$ 2 latent dimensionalities ($L = 10$ and $L = 50$) $\times$ 5 random seeds.

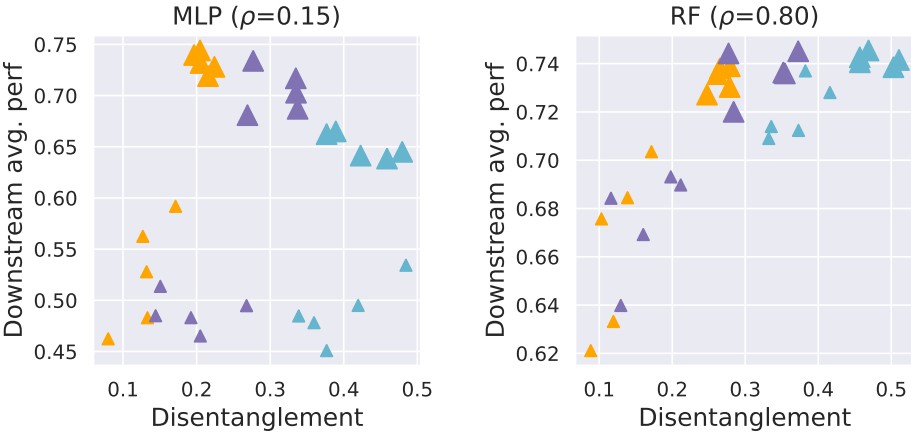

Figure 7: **Disentanglement (D) vs. downstream performance.** Scatter plots show 30 data points: 3 models (AEs, VAEs, $\beta$-VAEs) $\times$ 2 latent dimensionalities ($L = 10$ and $L = 50$) $\times$ 5 random seeds.

### B.2 DIFFERENT AMOUNTS OF DATA

In Fig. 10 we present loss-capacity curves obtained when using different amounts of data to train the MLP probes. As shown, larger datasets have smaller performance gaps between (i) synthetic and learned representations; and (ii) small and large representations.

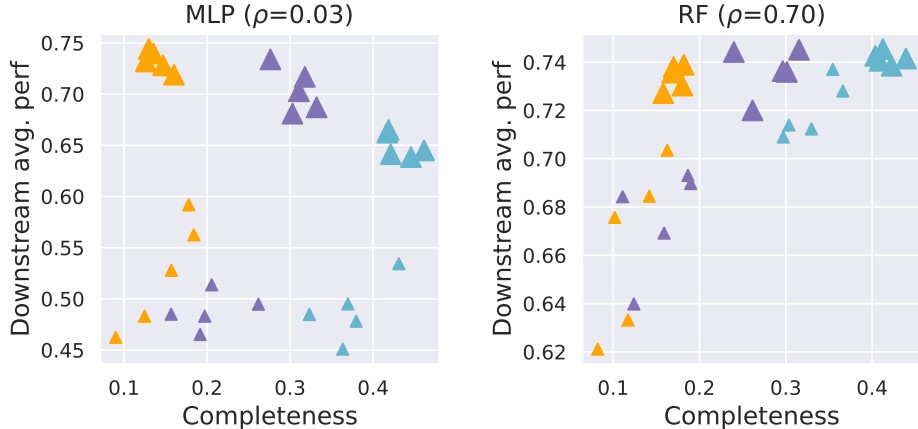

Figure 8: **Completeness (C) vs. downstream performance.** Scatter plots show 30 data points: 3 models (AEs, VAEs, $\beta$-VAEs) $\times$ 2 latent dimensionalities ($L = 10$ and $L = 50$) $\times$ 5 random seeds.

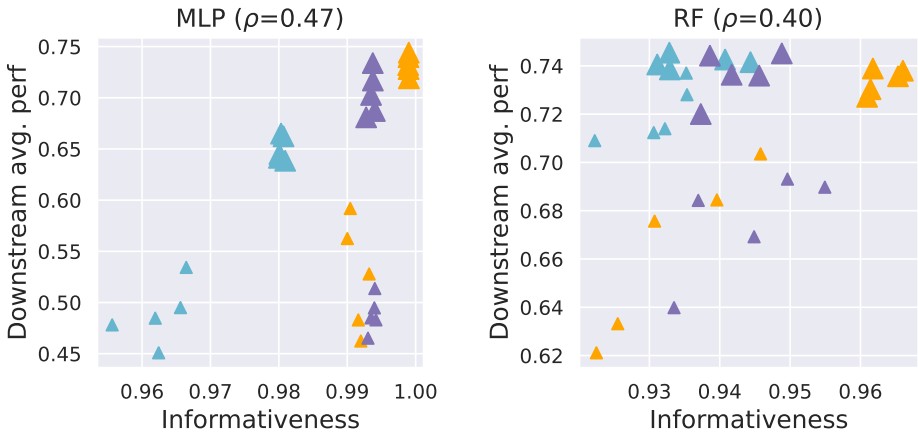

Figure 9: **Informativeness (I) vs. downstream performance.** Scatter plots show 30 data points: 3 models (AEs, VAEs, $\beta$-VAEs) $\times$ 2 latent dimensionalities ($L = 10$ and $L = 50$) $\times$ 5 random seeds.

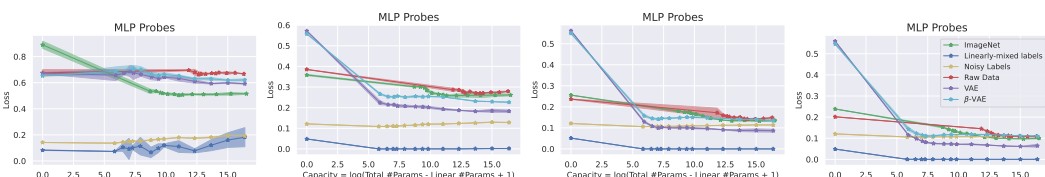

Figure 10: Loss-capacity curves for `MPI3D-Real` subsets of size 100, 1000, 5000 and 10000 respectively.