# OpenReview forum: "DCI-ES: An Extended Disentanglement Framework with Connections to Identifiability"
_ICLR.cc/2023/Conference — ICLR 2023 poster_

### Official Review · Reviewer_xBj7 · 2022-10-24

**Confidence:** 4
**Clarity, Quality, Novelty And Reproducibility:** Please check my review above
**Correctness:** 3
**Technical Novelty And Significance:** 2
**Empirical Novelty And Significance:** 2
**Recommendation:** 6

**Strength And Weaknesses:**

### Strengths
1) Overall, the writing of the paper is good

### Weaknesses
1) The novelty and significance of the work are low as it is mainly developed on an existing disentanglement metric which is DCI.
2) The DCI metric was proposed long ago and has several drawbacks such as i) only supporting continuous factors, ii) using a not-well-normalized importance matrix R, …  as discussed in a more recent work by Do & Tran ICLR-2020 [1] (Appendix A.8). The current work inherits all these drawbacks from DCI. It even complicates things by introducing 2 new minor metrics, while in my opinion, it should address the limitations of DCI and propose a simpler yet better metric for disentanglement.
3) Lack of discussion of several related works about metrics for disentanglement [1, 2].
4) Linking DCI to identifiability is straightforward and can be proven quite easily. Thus, Proposition 3.3 in the paper does not sound interesting to me. Besides, the DCI only achieves identifiability in the extreme case when D=C=1. But in practice, we rarely achieve such perfect values of D and C. Normally, D, and C will be between 0 and 1. How can the authors quantify the “amount” of identifiability in this case? I think answering this question is more worthwhile than introducing 2 complementary minor metrics.
5) The explicitness metric seems complex and ad-hoc to me. Since the capacity Cap(.) is defined differently for different kinds of models (e.g., Cap(.) is the maximum tree depth for a random forest while it is the number of neurons in neural networks), explicitness will be different if different models are used, which is not desirable. There is no consistent strategy to choose the “probe capacities” $\kappa\_1$, …, $\kappa\_T$. In addition, we need to train a new model for each “probe capacity” which can be very costly if the model size or the number of probe capacities is large.  The suitable base loss $\ell^b_j$ and the suitable lowest loss $\ell^{*}_j$ are not well defined and not consistent between models. I don’t understand what $\mathbb{E}[z_j]$ really means for $\ell^b_j$. Moreover, the loss $\ell^{t, c}$ for each probe capacity $\kappa_t$ can be noisy and incorrect, which greatly affects the accuracy of the metric. The explicitness E=1 means that AULCC=0, which is almost unlikely to happen.
6) I don’t see any point in using Size (S) as a metric.
7) The results in Table 2 are very hard to interpret as I don’t understand which representations are the best. Besides, there is no clear evaluation target for the representations. Looking at Table 2, how can we know the two proposed metrics E and S are correct and reasonable?

[1] Theory and Evaluation Metrics for Learning Disentangled Representations, Kien Do and Truyen Tran, ICLR-2020.

[2] Weakly Supervised Disentanglement with Guarantees, Shu et al., ICLR-2020.


**Summary Of The Paper:**

This paper connects the DCI framework [1] for disentanglement to identifiability and proposes an extension of the framework that incorporates two new measures of representation quality which are explicitness (E) and size (S).

**Summary Of The Review:**

This paper is well written. However, its novelty and significance are limited. It seems to complicate things by introducing two minor, ad-hoc metrics besides DCI instead of designing better, more elegant metrics. Thus, I think the paper is below the acceptance bar of ICLR.

### Post rebuttal
******************************************
I would like to thank the authors for your comments. I think most of my concerns are well addressed by the authors. Thus, I raise my score to 6 though I still worry about the soundness and the correctness of the metric for practical use.

---

> ### Author Response · Authors · 2022-11-18
> **Response to Reviewer xBj7 (part 1)**
>
> We thank the reviewer for their time in reviewing our submission. Below we respond to each of the reviewer’s points. We hope that the reviewer can remain open in their evaluation of our paper, despite their strong initial opinions on the original DCI framework, and are happy to answer any further questions should they arise.
>
> > “The novelty and significance of the work are low as it is mainly developed on an existing disentanglement metric which is DCI.”
>
> Building on existing work is an integral part of the scientific process, not a reason for dismissal.
>
> > “The DCI metric was proposed long ago and has several drawbacks such as i) only supporting continuous factors, ii) using a not-well-normalized importance matrix R, … as discussed in a more recent work by Do & Tran ICLR-2020 [1] (Appendix A.8).”
>
> The DCI metric was proposed 4 years ago (2018) and remains a highly-cited, influential paper in the disentanglement community. **While we consider it out of scope to defend/write a rebuttal for the already published work of Eastwood & Williams (2018) upon which we build, we are nonetheless happy to engage and point out that the mentioned drawbacks are not true**. In particular, the DCI framework supports discrete factors and $R$ is indeed well normalised. With regard to Do & Tran (2020, Appendix 8), we find their three drawbacks inaccurate, as we will now detail:
> 1. *“First, using three different metrics to quantify disentangled representations is not as convenient as using a single metric like MIG”*. Reporting separate scores is not a weakness/drawback but rather a major strength—it permits a fine-grained comparison of representations. In particular, representations can be disentangled but not complete, informative but not disentangled, etc. Combining these separate measures into one discards this important fine-grained information. As explained in the original DCI paper, *“We report separate scores as they capture distinct criteria”*.
> 2. *“Second, these metrics do not apply for categorical factors with C classes since in this case the model weight is not a vector but an L × C matrix”*. This statement is incorrect. It is very easy, and arguably even more natural, to define a matrix of relative importances for categorical factors $z_j$. For example, by using Gini Importance with random-forest classifiers—as suggested in the original DCI paper.
> 3. *“Third, defining the pseudo-distribution $P_{ik}$ [...] seems ad hoc because i) the weight magnitudes $R_{ik}$ are unbounded and can vary significantly (see Appdx. A.9), and ii) $P_{ik}$ strongly depends on the available ground truth factors”*. With regard to (i), this statement is incorrect—$R_{ik}$ *is* indeed bounded since the columns of $R$ are non-negative and sum to one. While this may not have been perfectly clear in the original DCI paper, it is in our Definition 2.1 and our proof of Corollary 3.4 in Appendix A.2. With regard to (ii), we do not understand why this is a drawback—the scores rightly depend on *all* ground-truth factors.
>
>
> > “introducing 2 new minor metrics, while in my opinion, it should address the limitations of DCI and propose a simpler yet better metric for disentanglement.”
>
> We respectfully disagree with the reviewer’s opinionated comment. As argued in detail above, the reviewer’s arguments against the original DCI framework are factually incorrect. We actually do consider DCI-ES to be a better, more refined version of DCI, with E a fundamentally new insight / notion of disentanglement rather than a mere “minor metric”. With regard to size (S), as pointed out by Reviewer wQrr, *“The ‘size’ metric, although very simple and obvious, is important if one wants to make fair comparisons”*.
>
> > “Lack of discussion of several related works about metrics for disentanglement [1, 2]”
>
> While we already defer to Locatello et al. (Section 6, 2020) for an in-depth discussion of different disentanglement metrics and their pros/cons, we have added these two related works/metrics to the eight already mentioned in the introduction.
>
> > “Linking DCI to identifiability is straightforward and can be proven quite easily. Thus, Proposition 3.3 in the paper does not sound interesting to me.”
>
> Although the results proved are indeed relatively straightforward, these connections were not clearly presented before, which, as pointed out by Reviewer wQrr, makes the results *"novel"* and a *"valuable contribution"* (in connecting two communities).

---

> > ### Author Response · Authors · 2022-11-18
> > **Response to Reviewer xBj7 (part 2)**
> >
> > > “Besides, the DCI only achieves identifiability in the extreme case when D=C=1. But in practice, we rarely achieve such perfect values of D and C. Normally, D, and C will be between 0 and 1. How can the authors quantify the “amount” of identifiability in this case? I think answering this question is more worthwhile than introducing 2 complementary minor metrics.”
> >
> > As detailed in the penultimate paragraph of Section 4.2, our framework does indeed provide a fine-grained picture of identifiability. In particular, when D=C=1 does not hold, we recover weaker forms of identifiability.
> >
> > >“The explicitness metric seems complex and ad-hoc to me. Since the capacity Cap(.) is defined differently for different kinds of models (e.g., Cap(.) is the maximum tree depth for a random forest while it is the number of neurons in neural networks), explicitness will be different if different models are used, which is not desirable.”
> >
> > We are unsure whether we understand the reviewers point here. Different probes or function classes naturally have different measures of capacity. This is not an issue since all representations are compared using the same probe class. Furthermore, in practice, one can simply choose the probe class which will be used downstream. In our work, we used three different probe classes simply to illustrate that the loss-capacity relationship or explicitness notion holds for different probe classes—e.g. Fig. 1 shows similar trends for the three probe classes.
> >
> > > “There is no consistent strategy to choose the ‘probe capacities’”
> >
> > This is not true—we suggest a strategy in Section 4.1.
> >
> > > “In addition, we need to train a new model for each “probe capacity” which can be very costly if the model size or the number of probe capacities is large.”
> >
> > We agree that this could be costly if very large probes were needed to predict $\mathbf{z}$ from $\mathbf{c}$. However, this is rarely the case, particularly for the synthetic datasets on which disentanglement methods are usually evaluated.
> >
> > > “The suitable base loss $\ell_j^b$ and the suitable lowest loss $\ell_j^*$ are not well defined and not consistent between models.”
> >
> > This is again not true—they are indeed well defined for each probe class and shared across representations, making the comparisons consistent—see Section 4.1. Furthermore, as it seems to be implied by the reviewer here again, we reiterate that it does not make sense for either the capacities or losses to be consistent across probes or function classes—each has its own notion of capacity and will naturally achieve different performances, e.g. the maximum tree depth for random forests or the number of basis functions for random Fourier features. **What matters is that we can compare representations consistently for a given probe class, which we can**.
> >
> >
> > > “I don’t understand what $\mathbb{E}[z_j]$ really means for $\ell_j^b$ ”
> >
> > This means that the mean of $z_j$ is used as a (constant) baseline predictor.
> >
> > > “Moreover, the loss $\ell^{t,c}$ for each probe capacity $\kappa_t$ can be noisy and incorrect, which greatly affects the accuracy of the metric.”
> >
> > Noisy loss-estimates are unfortunately an unavoidable consequence of finite-data evaluations. We do not see this as a limitation, since it applies to almost all of machine learning and statistics.
> >
> > > “The explicitness $E=1$ means that $AULCC=0$, which is almost unlikely to happen.”
> >
> > We do not understand why this would be a problem; perhaps the reviewer can clarify.
> >
> > > “I don’t see any point in using Size (S) as a metric”
> >
> > As pointed out by Reviewer wQrr: *"The ‘size’ metric, although very simple and obvious, is important if one wants to make fair comparisons"*. Moreover, as explained in the first paragraph of Section 4.2 and demonstrated in both Table 1 and Figure 3, our size metric is *“motivated by the observation that larger representations tend to be both more informative and more explicit (see Tab. 1 [...]). Reporting S thus allows size-informativeness and size-explicitness trade-offs to be analysed”*.

---

> > > ### Author Response · Authors · 2022-11-18
> > > **Response to Reviewer xBj7 (part 3)**
> > >
> > > > “The results in Table 2 are very hard to interpret as I don’t understand which representations are the best. Besides, there is no clear evaluation target for the representations. Looking at Table 2, how can we know the two proposed metrics E and S are correct and reasonable?”
> > >
> > > **The main insight of this table, and indeed of the entire paper and framework, is that no single representation is “best” along all dimensions or in terms of all metrics**. Some representations are more disentangled, some more complete, some more informative, some more explicit, some smaller. While the reviewer seems keen on reducing the evaluation of representations to a single score—as per the referenced comment (1) of Do & Tran (2020, Appendix 8) above—we strongly advise against such a reductionist approach, even if it seems “convenient”. **We report 5 distinct scores (DCIES) as they capture 5 distinct criteria that one may seek when learning representations. Which metrics are deemed most important, and which representations are deemed “best”, ultimately depends on the application of interest.**
> > >
> > > We have updated the caption of Table 2 to make clear this important point.

---

> ### Comment · Reviewer_xBj7 · 2022-11-25
> **Response to the authors**
>
> After reading the authors’ rebuttal, I still find most of my concerns remain unaddressed. It seems that the authors didn’t really appreciate the true points in my comments. I would like to take some examples below.
>
> > The novelty and significance of the work are low as it is **mainly** developed on an existing disentanglement metric which is DCI.
>
> In this question, my concern is about novelty and significance, and I expect the authors to show that **in which ways** their method is novel and significant. The authors, instead of making this point clear, complain that this is not a reason for dismissal. The truth is, I haven’t made any decision yet and if I do, novelty and significance are just parts of it besides many other reasons.
>
>
> > “The DCI metric was proposed long ago and has several drawbacks such as i) only supporting continuous factors, ii) using a not-well-normalized importance matrix R, … as discussed in a more recent work by Do & Tran ICLR-2020 [1] (Appendix A.8).” The current work inherits all these drawbacks from DCI.
>
> Here, my concern is stated in the second sentence _“The current work inherits all these drawbacks from DCI”_, not the first sentence. I expect the authors to show me that _how their new metrics could mitigate the drawbacks of DCI_ (which are pointed out in another work or if they are aware of any). However, the authors ignore this sentence and are so passionate about defending DCI (which is **not** their proposed method). They don’t accept the fact that DCI **can** have drawbacks (as no method is totally perfect) and their method **can** inherit all these drawbacks.
>
> Regarding the two drawbacks that I took as examples from the work of Do & Tran, after carefully reading the authors’ arguments, the work of Do & Tran, and the original DCI paper, I feel that the drawbacks mentioned in the work of Do & Tran are mainly developed based on their experiments with LASSO. I agree with the authors that continuous factors and normalized matrix R are supported by using Random Forests. Nevertheless, I am also worried that do we have other choices for computing R apart from Random Forests without facing these drawbacks.
>
>
> > “Besides, the DCI only achieves identifiability in the extreme case when D=C=1. But in practice, we rarely achieve such perfect values of D and C. Normally, D, and C will be between 0 and 1. How can the authors quantify the “amount” of identifiability in this case? I think answering this question is more worthwhile than introducing 2 complementary minor metrics.”
>
> I just see in Section 4.1 (not 4.2) the authors say that _“Thus, if D = C = I = E = 1 does not hold exactly, which score deviates the most from one may provide valuable insight into the type of identifiability violation.”_ . In fact, if D, C, I are between [0, 1], they can give you a sense of the type and the amount of **disentanglement** being violated, not necessarily the _“amount of identifiability”_. How can you mathematically quantify the _“amount of identifiability”_ with D, C, I, E?
>
>
> > There is no consistent strategy to choose the ‘probe capacities’
>
> In Section 4.1, the authors only mention choosing probe capacities for Random Forests but not for other probe classes like MLPs. I still don’t get how the probe capacities are chosen for MLPs? Even for Random Forests, what will happen if $\kappa_1 \neq 1$  and $T$ is arbitrary?
>
>
> > I don’t understand what $\mathbb{E}[z_j]$ really means for $\ell^b_j$
>
> What I mean here is that $\mathbb{E}[z_j]$ is not a math form of a scalar loss but is used as a scalar loss. What will happen if $z_j$ is a categorical factor since in this case, $\mathbb{E}[z_j]$ is a vector of length $K$ not a scalar anymore?
>
>
> > Moreover, the loss $\ell^{t, c}$ for each probe capacity can be noisy and incorrect, which greatly affects the accuracy of the metric.
>
> I do not agree with the authors’ answer about this. Noise can be fine during training since it can improve generalization in some cases. However, during evaluation, noise should not happen as it can lead to inaccurate judgments.
>
>
> > The explicitness E=1 means that AULCC=0, which is almost unlikely to happen
>
> This is because, at AULCC=0, $\kappa_1 = 1$ and we can hardly achieve perfect classification/regression with a single-depth random forest.
>
> I also note that depth is just *one* hyperparameter of Random Forest besides various hyperparameters (e.g., min_samples_split, min_samples_leaf, min_weight_fraction_leaf, ...) which can be found at https://scikit-learn.org/stable/modules/generated/sklearn.ensemble.RandomForestClassifier.html . I am concerned that if different methods use the _same_ depth but _different_ values for other hyperparameters during evaluation, it may lead to unfair comparison.
>
> For the above reasons, I decided to keep my score unchanged.

---

> > ### Author Response · Authors · 2022-11-26
> > **Response to Reviewer xBj7**
> >
> > We are disappointed to read that the reviewer: (1) felt that some of their points were not addressed---we try to further clarify below; (2) has chosen to keep both their low score and maximum confidence---we believe that many points were addressed and many were shown to be false or incorrect; and (3) continues to focus on possible but now unnamed drawbacks of the previous DCI framework---after their initial drawbacks were shown to be incorrect.
> >
> > > The novelty and significance of the work are low **as** it is mainly developed on an existing disentanglement metric which is DCI.
> >
> > Here we took/take issue with the **reason** for dismissal or asserting that the novelty and significance are low. The reviewer clearly claims that the novelty and significance are low **because** it is mainly developed upon existing work.
> >
> > > They don’t accept the fact that DCI can have drawbacks (as no method is totally perfect) and their method can inherit all these drawbacks.
> >
> > We never claimed that the DCI framework of Eastwood & Williams (2018) does not have drawbacks, nor should we need to. We simply stated that the three alleged drawbacks of the reviewer were false, since they were being used as a reason to reject our paper / extended framework. Now that this has been cleared up, we're not sure of what drawbacks the reviewer is asking us to mitigate, since none have been named.
> >
> > > I am also worried that do we have other choices for computing $R$ apart from Random Forests without facing these drawbacks.
> >
> > In showing that the alleged drawbacks were false, we spoke of predictor-agnostic matrices $R$, so the reviewer need not be worried! Random-forest classifiers were only used afterwards as an example: *"For example, ..."*.
> >
> >
> > > In fact, if D, C, I are between [0, 1], they can give you a sense of the type and the amount of disentanglement being violated, not necessarily the “amount of identifiability”. How can you mathematically quantify the “amount of identifiability” with D, C, I, E?
> >
> > There seems to remain a misunderstanding here about what identifiability means, and how this concept differs from the empirical scores:
> > - **Identifiability** is a theoretical concept about whether certain learning tasks are feasible at all, given certain assumptions and infinite data. In practice, while this is not attainable, it can still be insightful to quantify how far one is from full identification. This motivation is also shared by other identifiability metrics, such as the mean correlation coefficient, or Amari distance. As detailed in the "A fine-grained picture of identifiability" paragraph towards the end of Section 4.1, our framework does indeed give insight into the type of identifiability violation, with different scores giving rise to different forms of identifiability. In particular, when D=C=1 does not hold, we recover weaker forms of identifiability.
> > - The **empirical scores** DCIES are all in [0, 1] and empirically measure of the quality of the representation in terms of a particular property.
> >
> >
> > > What I mean here is that $\mathbb{E}[z_j]$ is not a math form of a scalar loss but is used as a scalar loss. What will happen if $z_j$ is a categorical factor since in this case, $\mathbb{E}[z_j]$ is a vector of length $K$ not a scalar anymore?
> >
> > $\mathbb{E}[z_j]$ is used as a (constant) baseline predictor, not as the loss. The loss for example $i$ is: $\ell(z^i_j, \mathbb{E}[z_j])$.
> >
> >
> > > Noise can be fine during training since it can improve generalization in some cases. However, during evaluation, noise should not happen as it can lead to inaccurate judgments.
> >
> > We reiterate that noisy losses, whether during training or evaluation, are unavoidable with finite data.
> >
> > > I also note that depth is just one hyperparameter of Random Forest besides various hyperparameters (e.g., min_samples_split, min_samples_leaf, min_weight_fraction_leaf, ...) which can be found at https://scikit-learn.org/stable/modules/generated/sklearn.ensemble.RandomForestClassifier.html. I am concerned that if different methods use the same depth but different values for other hyperparameters during evaluation, it may lead to unfair comparison.
> >
> > The reviewer need not be concerned here as these additional hyperparamters are all: (1) fixed/shared across representations, making comparisons fair; (2) implicitly controlled by maximum depth.

---

### Official Review · Reviewer_wQrr · 2022-10-25

**Confidence:** 4
**Correctness:** 3
**Technical Novelty And Significance:** 2
**Empirical Novelty And Significance:** 3
**Recommendation:** 6

**Clarity, Quality, Novelty And Reproducibility:**

Lack of mathematical rigor:
- Corollary 3.4: In general, it is not clear that |W| qualifies as a valid choice of R (definition 2.1), since its columns might not sum to one. This means one cannot apply Prop. 3.3, which leverages the fact that the columns of R sum to one. I guess this could be fixed by redefining R by normalizing its entries to make sure its columns sum to one. But one has to be careful when doing that, since, a priori, |W| might have a column filled with zeros (this possibility isn’t excluded by the assumptions of the theorem, but that would imply some z_i = 0), which would prevent such a normalization. Also the proof mentions W^{-T}, but a priori it is not guaranteed that W is invertible (maybe this could be added to the assumption? That would also prevent one of the columns to be filled with zeros, thus allowing normalization). These subtleties must be addressed.
- Corollary 3.5: Please add the assumption that f is differentiable, since the statement refers to its partial derivative.
- (Not really about rigor, but a related point) I believe Corollary 3.5 would benefit from an example of R that satisfies this property (and nonlinear f). The points I raised about Corollary 3.4 shows that constructing a valid R can present subtleties. The following remark mentions the Gini importance for Random forest, but, as the authors acknowledge, isn’t invertible.

Clarity:
- As I said, this paper is overall well written and clarity is not an issue.
- In introduction: It is unclear what is meant by "a uniformly-mixed version thereof" (point (iii)). I find the explanation confusing and imprecise: "each ci containing the same amount of information about each zj". A solution would be to refer to the definition in Section 6.1.
- Section 4.1: “For example, we may choose κ_T to be large enough for all representations to achieve their lowest loss and, for random forest f’s, we may choose κ_1 = 1…” Does k_1 = 1 mean tree depth = 1? Say it explicitly here.

Novelty/Originality:
- Although the results proved are close to being trivial, these connections were not clearly presented before, which makes this work novel.
- The originality is somewhere between low and average. But this is still a valuable contribution.

Suggestions for improvement:
- Could Corollary 3.4 be seen as a special case of Corollary 3.5? For instance, could one present only Corollary 3.5 and give the linear case as an example?


**Strength And Weaknesses:**

Strengths:
- I believe both the deep learning community and the (more theoretically oriented) nonlinear ICA community will benefit from the connections made in this work.
- I very much agree with the authors that one should take into account the capacity of a probe when evaluating a representation and that exploring different classes of probes with different capacities gives a more complete picture of the learned representation.
- The "size" metric, although very simple and obvious, is important if one wants to make fair comparisons.
- The paper is clearly written and easy to follow.
- The experiments are clearly presented and sufficient for the argument.

Weaknesses:
- Mathematical precision is sometimes lacking. See my comment below, especially about Corollary 3.4.
- As I said, I agree with the author that one should take into account the capacity of a probe when evaluating a representation and that exploring different classes of probes with different capacities gives a more complete picture of the learned representation. However, I am a bit skeptical that the proposed explicitness score (E) brings more value than just transparently reporting the Informativeness score (I) for a couple of different probes with different capacities (i.e. the loss-capacity curves). E is basically a less transparent way of communicating this information. Moreover, Figure 4 shows that the ranking can change when simple rescaling are applied to the capacity measures. I find this worrying.
- I do not understand why the DCI metrics are restricted to R matrices with columns that sum to one (Definition 2.1). This is counterintuitive to me, since one could have a learned representation c that is completely useless to predict one of the ground-truth latent variable z_j, but since the columns of R must sum to one, it is impossible for the corresponding column (R_{.,j} to be filled with zeros (as it should, to represent the fact that the representation is useless to predict that ground-truth factor).

**Summary Of The Paper:**

This work draws connections between the DCI metrics, commonly used in the deep learning literature on disentanglement, and notions of representation identifiability, commonly used in the literature on independent component analysis (ICA). More precisely, Corollary 3.5 provides conditions under which D=C=1 and K=L implies equivalence between c and z, up to permutation and element-wise reparameterization. In addition, the DCI metrics are extended by adding "explicitness" (E) and "size" (S). Basically, the former measures how a given probe class can predict the latent variables with various capacity levels and the latter is simply the ratio of the dimensionality of the ground-truth latent vector over the dimensionality of the learned representation. These novel metrics are then evaluated on various learned representation and dataset.

**Summary Of The Review:**

I believe this paper makes valuable contributions, but (i) Corollary 3.4 contains a mistake (although I am optimistic that the issue can be resolved) and (ii) I do not see the added value in using E instead of simply reporting the loss-capacity curves more transparently. Mainly for these reasons, I recommend weak rejection for now, but I am very open to increasing it to a weak accept once the problem in Corollary 3.4 is fixed.

--- Post-rebuttal update ---
I have updated my score to 6, see my comment below.

---

> ### Author Response · Authors · 2022-11-18
> **Response to Reviewer wQrr (part 1)**
>
> We sincerely thank the reviewer for their careful reading and detailed, constructive comments. We respond to each point below.
>
>
> *Weaknesses:*
>
> > “I am a bit skeptical that the proposed explicitness score (E) brings more value than just transparently reporting the Informativeness score (I) for a couple of different probes with different capacities (i.e. the loss-capacity curves). E is basically a less transparent way of communicating this information.”
>
> We agree with the reviewer’s point that the explicitness score—a single-number summary of the loss-capacity curve—is less informative than the entire loss-capacity curve (or multiple points along the curve). However, it is not clear how one would compare the curves of different representations without some sort of summarization, and we believe explicitnesses to be a sensible such summarization. **E thus brings more value than the curves alone by allowing an easier single-number comparison of representations’ loss-capacity relationships (e.g. in a table). Furthermore, our newly added experiments demonstrate the value of reporting the summary score E through its strong correlation with downstream performance** when using low-capacity probes—see Section 6.3 of the updated manuscript and our response to Reviewer R81A (second point).
>
> > “Moreover, Figure 4 shows that the ranking can change when simple rescaling are applied to the capacity measures. I find this worrying.”
>
> While we agree that this ranking-change is not ideal, we note that: (i) **the major ranking change occurs for MLP probes and the raw-data representation** (an extreme representation used here for illustration, but rarely in practice when comparing learned representations); (ii) **the ranking for other probes and representations is mostly consistent across capacity scalings**; and (iii) **we included this figure to illustrate that measuring the capacity of MLPs is still an open problem**, with future advances likely to benefit our explicitness measure and clear-up these issues.
>
>
> > “I do not understand why the DCI metrics are restricted to R matrices with columns that sum to one (Definition 2.1). This is counterintuitive to me, since one could have a learned representation c that is completely useless to predict one of the ground-truth latent variable z_j, but since the columns of R must sum to one, it is impossible for the corresponding column (R_{.,j} to be filled with zeros (as it should, to represent the fact that the representation is useless to predict that ground-truth factor).”
>
> To see why this is the case, it is important to note that the DCI framework defines disentanglement and completeness as **relative** concepts, with disentanglement capturing the relative contribution of $c_i$ to the prediction of each $z_j$, and completeness the relative contribution of each $c_i$ to a $z_j$. **The absolute usefulness of $\mathbf{c}$ for predicting $z_j$ is captured by the informativeness score $I_j$**, which can of course be 0. The DCI metrics thus capture both the *relative* mixing between the dimensions of $\mathbf{c}$ and $\mathbf{z}$ (D,C) as well as the *absolute* usefulness of $\mathbf{c}$ for predicting each $z_j$ ($I_j$). We believe that this separation of relative "mixing" and absolute informativeness to be correct, leading to 3 distinct measures which capture 3 distinct criteria. For example, if we add random noise to $z_j$, $I_j$ decreases but the D and C scores remain unchanged—we believe this to be best.

---

> > ### Author Response · Authors · 2022-11-18
> > **Response to Reviewer wQrr (part 2)**
> >
> > *Clarity, Quality, Novelty And Reproducibility:*
> >
> > > “Corollary 3.4: In general, it is not clear that |W| qualifies as a valid choice of R (definition 2.1), since its columns might not sum to one. This means one cannot apply Prop. 3.3, which leverages the fact that the columns of R sum to one. I guess this could be fixed by redefining R by normalizing its entries to make sure its columns sum to one. But one has to be careful when doing that, since, a priori, |W| might have a column filled with zeros (this possibility isn’t excluded by the assumptions of the theorem, but that would imply some z_i = 0), which would prevent such a normalization. Also the proof mentions W^{-T}, but a priori it is not guaranteed that W is invertible (maybe this could be added to the assumption? That would also prevent one of the columns to be filled with zeros, thus allowing normalization). These subtleties must be addressed.”
> >
> > Thank you for the careful checking and for bringing this to our attention. We initially thought that this directly follows from $R=|W|$---the choice of Eastwood & Williams (2018; Section 4.3) for LASSO-based regression, albeit with a different normalisation of $R$---in combination with the normalisation of $\mathbf{c}$ and $\mathbf{z}$ (see footnote 1). However, you are indeed right that this argument requires a separate normalisation of the columns. As stated, the result still holds with this adjusted definition of $R$, but requires a longer proof which, for space reasons, we now provide in Appendix A (along with all other proofs).
> >
> > In short, a zero column of $|W|$ is ruled out as this would imply that one of the $z_j$ has zero variance. This can be used to show that a column-normalised version of $|W|$ is indeed a valid choice for $R$, and that $W$ must hence be a generalised permutation matrix. This in turn implies that it has an inverse (so this actually need not be explicitly assumed *apriori*), and using the normalisation of variance 1 from footnote 1, we can finally show that $\mathbf{c}$ and $\mathbf{z}$ are related by a signed permutation matrix.
> >
> >
> > > “Corollary 3.5: Please add the assumption that f is differentiable, since the statement refers to its partial derivative.”
> >
> > Thank you for spotting this, we have added the differentiability assumption.
> >
> > > “(Not really about rigor, but a related point) I believe Corollary 3.5 would benefit from an example of R that satisfies this property (and nonlinear f). The points I raised about Corollary 3.4 shows that constructing a valid R can present subtleties. The following remark mentions the Gini importance for Random forest, but, as the authors acknowledge, isn’t invertible.”
> >
> > We agree with the reviewer here that such an example would be beneficial. While SAGE feature importances also do not satisfy Corrolory 3.5, in Section 4.3 we hypothesize that alternative methods which look at a feature’s mean *absolute* attribution value (Lundberg & Lee, 2017) may satisfy this property, the intuition being that absolute contributions do not allow for the cancellation of positive and negative contributions within the average; we have added this intuition to Section 4.3.
> >
> >
> > > “In introduction: It is unclear what is meant by "a uniformly-mixed version thereof" (point (iii)). I find the explanation confusing and imprecise: "each ci containing the same amount of information about each zj". A solution would be to refer to the definition in Section 6.1.”
> >
> > We have now clarified this sentence in the introduction, adding a pointer to Section 6.1 for the precise definition of the “uniformly-mixed” representation.
> >
> >
> > > “Section 4.1: “For example, we may choose κ_T to be large enough for all representations to achieve their lowest loss and, for random forest f’s, we may choose κ_1 = 1…” Does k_1 = 1 mean tree depth = 1? Say it explicitly here.”
> >
> > Yes this means a maximum tree depth = 1; we have now added this clarification.
> >
> > > “Could Corollary 3.4 be seen as a special case of Corollary 3.5? For instance, could one present only Corollary 3.5 and give the linear case as an example?”
> >
> > Thank you for the suggestion. Indeed, $R$, as defined in Corollary 3.4, satisfies the requirements of Corollary 3.5: in particular the main assumption that $R_{ij}=0$ iff. $f_j$ does not depend on $c_i$. However, the identifiability type of Corollary 3.4 (up to sign and permutation) is much stronger than that of Corollary 3.5 (element-wise reparametrisation). While we acknowledge that Corollary 3.4 may probably also be derived from Corollary 3.5 by leveraging the additional linearity assumption on $f$, the updated form of Corollary 3.4 with column-normalisation also informs us of how to construct an appropriate $R$ in the case of linear regression—instead of directly assuming that some valid $R$ is given. For these reasons, we consider it appropriate to present Corollary 3.4 as its own separate result.

---

> > > ### Comment · Reviewer_wQrr · 2022-11-23
> > > **Response**
> > >
> > > I would like to thank the authors for engaging with the points I raised in my review. I find most answers satisfying. I will increase my score to 6, as my main concerns have been addressed. I won't increase my score further because I still believe there is some arbitrariness in the specific way the metric E is computed, which I would like to see addressed based on theoretical principles instead of empirical evaluation (as the one provided in the updated manuscript).
> > >
> > > **Corollary 3.4:**
> > >
> > > Looks good to me. However, I would add explicitly in the statement of the corollary that z and c are standardized to have mean zero and variance one. I know this was mentioned in the text, but given how crucial this subtle point is to the argument, it should be stated.

---

> > > > ### Author Response · Authors · 2022-11-24
> > > > **Response to Reviewer wQrr**
> > > >
> > > > We would like to thank the reviewer for their continued engagement and for updating their score.
> > > >
> > > >
> > > > We will explicitly state that z and c are standardized in the statement of the corollary, as suggested.

---

### Official Review · Reviewer_WGrh · 2022-10-27

**Confidence:** 3
**Clarity, Quality, Novelty And Reproducibility:** See above
**Correctness:** 3
**Technical Novelty And Significance:** 3
**Empirical Novelty And Significance:** 2
**Recommendation:** 6

**Strength And Weaknesses:**

**Strength**

1 The new measurement of explicitness (E) (easy-to-use) and size (E) is interesting for disentanglement evaluation.

2 The paper writing is clear and easy to follow.

**Weaknesses**

1 The traditional DCI framework may already be considered explicitness(E) and size(S). For instance, to evaluate the disentanglement (D) of different representation methods, you may need to use a fixed capacity of probing (f), and the latent size should also be fixed. DCI and ES may be entangled with each other. For instance, if you change the capacity of probing or the latent size, then the DCI evaluation also changes correspondingly. The reviewer still needs clarification on the motivation for considering explicitness(E) and size(S) as extra evaluation.

2 Intuitively, explicitness(E) and size(S) may be highly related to the given dataset. The different capacity requirements in the 3rd paragraph may be due to the input modality difference. Given a fixed dataset, the evaluation of disentanglement should provide enough capacity and training time which is powerful enough to achieve the DCI evaluation. If the capacity of probing needs to be evaluated, then the training time, cost, and learning rate may also be considered because they may influence the final value of DCI.

**Summary Of The Paper:**

This paper proposes an extended Disentanglement Framework to evaluate the disentanglement in representation learning. Specifically, the paper first links the disentanglement to independent component analysis, then propose two new measures of representation quality: explicitness (E) and size (S). The paper uses MPI3D and Cars3D datasets to evaluate the method.


**Summary Of The Review:**

The reviewer thinks the disentanglement evaluation should consider relatively orthogonal and important evaluation metrics. Motivation and necessity are the primary concerns.



--------------------------------------------------------
Thanks for the author's feedback! After reading the feedback, some concerns are addressed, e.g., the motivation for measuring explicitness(E) and size(S). While the reviewer's second question was not well addressed, "If the capacity of probing needs to be evaluated, then the training time, cost, and learning rate may also be considered because they may influence the final value of DCI.". After reading other reviewers' comments, the reviewer has similar concerns about the proposed method's novelty (technical contribution) compared to DCI and practical usefulness. Therefore, the reviewer will keep the original score and vote for borderline.

---

> ### Author Response · Authors · 2022-11-18
> **Response to Reviewer WGrh (part 1)**
>
> We thank the reviewer for their constructive comments. We respond to each point below.
>
> *Weaknesses:*
>
> > “The traditional DCI framework may already be considered explicitness(E) and size(S). For instance, to evaluate the disentanglement (D) of different representation methods, you may need to use a fixed capacity of probing (f), and the latent size should also be fixed. DCI and ES may be entangled with each other. For instance, if you change the capacity of probing or the latent size, then the DCI evaluation also changes correspondingly. The reviewer still needs clarification on the motivation for considering explicitness(E) and size(S) as extra evaluation.”
>
> **While the original DCI metrics may be _affected_ by capacity and size, they do not measure them**, meaning the explicitness and size of representations cannot be compared.
>
> In addition, the reviewer’s comment about using a fixed capacity and size seems to indicate some confusion about the framework, which we will now try to clarify. Firstly, we wish to compare **different representations** which may: (1) require different capacity probes to predict $\mathbf{z}$ from $\mathbf{c}$; and (2) be of different sizes. We are not evaluating a single representation. Secondly, explicitness or _ease-of-use_ is about how the loss/informativeness changes **as a function of capacity**—it is not about the performance with one particular fixed capacity. As in the related work section: *“the informativeness score with a linear probe quantifies the amount of information in c about z that is ``explicitly represented'', while [...] our DCI-ES framework differentiates between the amount of information in c about z (informativeness) and the ease-of-use of this information (explicitness). This allows a more fine-grained analysis of the relationship between c and z—both theoretically (distinguishing between more identifiability equivalence classes; Sec. 3) and empirically (comparing many different representations; Sec. 6)”*.
>
> Finally, the reviewer’s comments on the DCI scores and ES scores being entangled also indicates some confusion about the extended framework, and how it measures these *distinct* properties. We will now try to clarify. Firstly, it is true that the D, C and I metrics are _affected_ by the capacity of $f$, e.g. if we use linear vs nonlinear probes we may get different scores. However, **they should always be computed on the lowest-error or highest-informativeness probe in order to yield accurate scores, as the reviewer themselves note in their second comment below**. That is, D, C and I should have no dependence on the capacity of $f$ so long as $f$ has enough capacity to extract all of the information about $\mathbf{z}$ in $\mathbf{c}$ (in order to get the highest informativeness score possible with a given representation $\mathbf{c}$).
>
> In sum, so long as we use sufficient capacity to extract all of the information in $\mathbf{c}$ about $\mathbf{z}$, then the D, C and I scores are not dependent on the capacity of $f$. **This allows us to measure and compare the explicitness or ease-of-use of representations as a distinct property**. A similar argument follows for size (S), giving us 5 scores which capture 5 distinct properties, ultimately allowing us to compare representations along 5 distinct axes: D, C, I, E, and S. As pointed out by Reviewer wQrr, *“The ‘size’ metric, although very simple and obvious, is important if one wants to make fair comparisons”*. Moreover, as explained in the first paragraph of Section 4.2 and demonstrated in both Table 1 and Figure 3, our size metric was *“motivated by the observation that larger representations tend to be both more informative and more explicit (see Tab. 1 [...]). Reporting S thus allows size-informativeness and size-explicitness trade-offs to be analysed”*.
>
> We hope that the motivation and necessity for the extended DCI-ES framework are now clear. If not, we would be happy to answer any further questions. We end by noting that we have added further experiments to motivate the use of our E score—see Section 6.3 of the updated manuscript for details, as well as our response to Reviewer R81A (second point).

---

> > ### Author Response · Authors · 2022-11-18
> > **Response to Reviewer WGrh (part 2)**
> >
> > > “Intuitively, explicitness(E) and size(S) may be highly related to the given dataset. The different capacity requirements in the 3rd paragraph may be due to the input modality difference. Given a fixed dataset, the evaluation of disentanglement should provide enough capacity and training time which is powerful enough to achieve the DCI evaluation. If the capacity of probing needs to be evaluated, then the training time, cost, and learning rate may also be considered because they may influence the final value of DCI.”
> >
> > **Our extended framework, like the original DCI framework, provides a way in which to compare different representations on the same dataset**. For two different datasets and three different probes on each, Fig. 1 shows that different representations have different loss-capacity curves, resulting in different explicitness scores. However, the trends and insights provided by the DCIES scores remain consistent.
> >
> >
> > *Correctness:*
> >
> > > Several of the paper’s claims are incorrect or not well-supported.
> >
> > Which claims does the reviewer believe are incorrect or not well-supported? We did not see any mentioned/highlighted.

---

> ### Author Response · Authors · 2022-11-26
> **Reviewer WGrh: Has our response addressed your concerns?**
>
> Dear Reviewer WGrh,
>
> We would like to kindly ask if our response has addressed your concerns. If so, we ask that you consider adjusting your score. If not, we are happy to provide further clarifications. Thank you again for your time and continued engagement.
>
> Best regards,
>
> Authors

---

> ### Author Response · Authors · 2022-12-06
> **Response to Reviewer WGrh's post-rebuttal update**
>
> We thank the reviewer for their continued engagement during the discussion phase.
>
> > "Some concerns are addressed, e.g., the motivation for measuring explicitness(E) and size(S)."
>
> We are glad that the motivation is now clear.
>
> > "While the reviewer's second question was not well addressed, 'If the capacity of probing needs to be evaluated, then the training time, cost, and learning rate may also be considered because they may influence the final value of DCI.' "
>
> We apologise for failing to satisfactorily address this question before, and strive to do so now. First, we reiterate/clarify the evaluation process for the DCI and E scores:
>  1. Train T probes with increasing capacity, e.g. we could train T MLPs with 0, 1, …, T-1 hidden layers, each time sweeping over learning rates and choosing the best-performing one on a validation set.
>  2. Compute the DCI scores using the **single best-performing (i.e., most-informative) probe** (usually highest-capacity, single slice of the loss-capacity curve).
>  3. Compute the explicitness (E) score using the T probes of different capacities (area under the **entire loss-capacity curve**).
>
> With this evaluation procedure in mind, note that **probe capacity is the fundamental property of the $\mathbf{z}$-$\mathbf{c}$ relationship which we wish to measure**, as motivated in the paper from both theoretical (Section 3) and empirical (Section 6.3) perspectives. While training time and compute cost can be seen as hyperparameters regularising this capacity, they have **no influence on our evaluation or scores since we do not constrain the training time or compute cost**. As for the learning rate, we use a standard cross-validation procedure to select the best-performing value. While this may have some implicit influence on the **effective capacity** of the trained probe, this is not an issue for our evaluation since we focus on the **available capacity** of the probe class. In particular, **by increasing the available capacity in a controlled and systematic manner, we measure how the loss decreases as more capacity becomes available**.
>
> > "After reading other reviewers' comments, the reviewer has similar concerns about the proposed method's novelty (technical contribution) compared to DCI and practical usefulness. Therefore, the reviewer will keep the original score and vote for borderline."
>
> We would like to point out that, following the new experiments (Section 6.3), updates to the manuscript, and additional explanations/clarifications, **all other reviewers have updated their opinion and increased their score (some of whom previously had concerns about novelty and practical usefulness)**. In particular, Reviewer R81A, who's main concern was practical usefulness, has updated their score from 5 to 8, concluding that: *“The additional results and explanations make a lot of sense, and make the proposed method more appealing from my point of view”*.

---

> > ### Comment · Reviewer_WGrh · 2022-12-09
> > **Response to authors**
> >
> > Thanks for the author's feedback. It is helpful to understand more details about the choice of evaluation parameters. The reviewer will increase the score to 6 while still concerned about practical usage's high cost and complexity during the evaluation of DCI and E scores.

---

### Official Review · Reviewer_R81A · 2022-10-28

**Confidence:** 4
**Correctness:** 3
**Technical Novelty And Significance:** 3
**Empirical Novelty And Significance:** 2
**Recommendation:** 8

**Clarity, Quality, Novelty And Reproducibility:**

The paper is quite clear. This precise idea is novel as far as I know, but it of course builds upon and expands previous ideas on disentanglement. I'm fairly confident I could reproduce this work.

**Strength And Weaknesses:**

### Strengths
- The paper is certainly well written
- The work touches on an important open problem, how do we evaluate disentanglement?
- The proposed ideas are intuitively appealing

### Weaknesses
- The main proposed quantity, explicitness, relies heavily on other quantities for which it is still an open problem how to measure
- The empirical validation is a bit lacking, what are these measures predictive of? What are they a model of?


### Comments

> Note that $I_j$ depends on the capacity of $f_j$, as depicted in Fig. 1.

Shouldn't D and C also depend on the capacity of $f$ through $R$?

> Corollary 3.5, [..] with $f$ an invertible nonlinear function

This seems to be a really strong requirement, which greatly reduces the class of functions that the results of section 3 can give us intuition, or "theoretical insights", about. I wonder if I am missing something deeper here?


> Larger representations are often more informative. When dim$(c)<$dim$(z)$.

I know that this is just meant to be an intuitive passage, but I'm not sure this is the right threshold for dim$(c)$. Perhaps appealing to compression-based generalization theory here might make more sense, if the number of bits one can encode in $c$ is smaller than the number of bits of information in $z$, then there is necessarily degradation/lossy compression. Some phase transition occurs at that threshold, although as the authors already argue, regardless of the threshold more capacity is generally accepted to be more informative.


> larger representations (ImgNet-pretr, raw data) tend to be more explicit than smaller ones (VAE, $\beta$-VAE)

There's an experiment that's missing here, that might make this claim much easier to validate. What happens to a [$\beta$-]VAE as $L$ increases, i.e. S decreases?

The downside of asking this question is that, as the authors point out, changing $L$ changes the architecture of the model, which inherently changes the explicitness. Are two models with slightly different architectures comparable? Would it even make sense to plot E as a function of $L$?

As Figure 4 shows, even the choice of scaling on the AULCC affects the ranking of methods for which we have a poor measure of capacity, MLPs. This work leans heavily on a choice of quantification of capacity, which is still an open problem for DNNs.

It's obvious from recent research on generalization that we cannot hope to get interesting quantities that speak to generalization and disentanglement without taking into account (effective) capacity and data. This is why what is proposed is quite appealing intuitively. But I wonder what this quantity currently predicts, i.e. what it is a model of, what other quantities it correlates with and/or predicts.

In the face of this brittleness, one way to reassure ourselves is to gather more data. Perhaps such a plot as I'm proposing above would reveal some interesting (if noisy) patterns.

**Summary Of The Paper:**

This paper introduces two quantifiers of disentanglement (measures of how good a latent representation is, measurable when we have the ground truth latents), complementary to Eastwood & Williams' DCI, Disentanglement, Completeness, and Informativeness. The first is _explicitness_, which is related to a (normalized) area under the curve of informativeness as a function of capacity, and illustrates how accessible the true latents are from a code. The second is _size_, which simply is the ratio of the dimensionality of the code to the dimensionality of the true latents.

**Summary Of The Review:**

I'm leaning reject but still quite on the fence.

This paper provides a nice intuition and provides a decent amount of justification for it, but where the work is lacking is in showing where it leads. The authors propose a quantity, but it's not too clear how useful or predictive it is, nor if it has some empirical regularities which make it desirable in practice (if I wrote a paper on a new kind of VAE tomorrow, would I want to use this? I'm not convinced).

Update: The authors have addressed most of my concerns, and going through the updated version of the paper, I think it is now a much better work.

---

> ### Author Response · Authors · 2022-11-18
> **Response to Reviewer R81A (part 1)**
>
> We sincerely thank the reviewer for their thorough review and helpful feedback. We address the main concerns and questions below.
>
> _Weaknesses:_
>
> > “The main proposed quantity, explicitness, relies heavily on other quantities for which it is still an open problem how to measure”
>
> While we agree that measuring the capacity **of MLPs** is still an open problem, we do not believe this to be a major weakness for the following reasons:
> 1. **Our framework represents a first step for measuring explicitness**, ultimately presenting a fundamentally new notion and measure of "disentanglement". This highlights the importance of measuring the capacity required to use a representation (i.e. explicitness), and opens up future lines of research for the disentanglement community.
> 2. **As MLP capacity-measures improve, so too will our explicitness measure**. Measuring MLP capacity is its own line of research. As it matures and improves, so too will our explicitness measure for MLPs.
> 3. **It is easy to measure capacity for many probe/function classes, just not MLPs**, e.g. random forests or random Fourier features.
>
> > “The empirical validation is a bit lacking, what are these measures predictive of? What are they a model of? [...] if I wrote a paper on a new kind of VAE tomorrow, would I want to use this?”
>
> We take the reviewers point here and believe it is a valid question. Firstly, we would like to point out that **the new concept of explicitness carries value in itself**, opening up a new line or research or way of thinking about disentanglement—even if currently imperfect in practice. That said, we agree with the reviewer that such a concept is more useful if its measure is predictive of some important quantity in practice.
>
>
> **We have thus run additional experiments to investigate how predictive explicitness is of downstream performance when using low-capacity probes**. As detailed in Section 6.3 of the updated manuscript (see, e.g., Fig.4a) and summarised in the table below, explicitness better predicts downstream performance with low-capacity probes than any other measure (D, C or I). This is particularly true for MLP probes, where better “mixing-based” disentanglement (D,C) does not translate into better downstream performance (as shown by the very weak correlation), while better explicitness does. **These results demonstrate the value that E adds to the existing DCI framework, and ultimately the practical usefulness of reporting E when comparing/evaluating learned representations—with explicitness translating into downstream ease-of-use or capacity-efficient performance**.
>
> | *Probe* $f$ | *Task*  | *D* | *C* | *I* | *E*    |
> |--------------------|----------------|------------|------------|------------|---------------|
> |       MLP          | Regression     | 0.16       | 0.04       | 0.46       | **0.96** |
> |       MLP          | Classification | 0.14       | 0.00       | 0.48       | **0.96** |
> |       RF             | Regression     | 0.76       | 0.66       | 0.42       | **0.84** |
> |       RF             | Classification | 0.81       | 0.72       | 0.35       | **0.89** |

---

> > ### Author Response · Authors · 2022-11-18
> > **Response to Reviewer R81A (part 2)**
> >
> > *Comments:*
> >
> > > "Shouldn't D and C also depend on the capacity of $f$?"
> >
> > While D and C change when we use different- or restricted-capacity $f$’s, they should always be computed on the lowest-error or highest-informativeness probe in order to yield accurate scores. That is, D and C have no dependence on the capacity of $f$ so long as $f$ has enough capacity to extract all of the information about $\mathbf{z}$ in $\mathbf{c}$ (in order to get the highest informativeness score possible with that given representation $\mathbf{c}$).
> >
> >
> > > “[The invertibility of $f$ in Corollary 3.5] seems to be a really strong requirement, which greatly reduces the class of functions that the results of section 3 can give us intuition, or ‘theoretical insights’, about. I wonder if I am missing something deeper here?”
> >
> > While we agree that it is indeed a rather strong requirement, it is a *necessary* one for drawing connections to identifiability. In particular, the strongest notion of identifiability refers to strict equality (i.e., $f$ being the identity function which is invertible); weaker notions of identifiability are expressed in terms of equivalence classes, and invertibility then follows from the symmetry property of equivalence relations. Intuitively, identifiability means that certain properties are preserved when going back and forth between two entities or variables, and this requires/necessitates that such a mapping be invertible.
> >
> >
> > Note, however, that identifiability is more of a theoretical guidance as to whether certain learning tasks are feasible at all, given certain assumptions and infinite data. In practice, while this is not attainable, it can still be insightful to quantify how far one is from full identification. This motivation is also shared by other identifiability metrics, such as the mean correlation coefficient, or Amari distance.
> >
> >
> > > Better to speak of information capacity than dimensionality in the intuitive passage about dim(c) < dim(z).
> >
> > While this was indeed more of an intuitive statement, with the dimensionality of continuous $\mathbf{c}$ and $\mathbf{z}$ omitting a close connection to their information-encoding capacities, we tend to agree with the reviewer that it may be more precise and general to speak of the number of bits that can be encoded in $\mathbf{c}$ and $\mathbf{z}$. We will update this passage in the next version of the manuscript.
> >
> >
> > > “There's an experiment that's missing here, that might make this claim much easier to validate. What happens to a $\beta$-VAE as $L$ increases, i.e. $S$ decreases?”
> >
> > We already conducted this experiment with an autoencoder (AE) – see Table 1 and Figure 3.
> >
> >
> > > “Are two models with slightly different architectures comparable? Would it even make sense to plot E as a function of $L$?”
> >
> > While we see the reviewer’s point here, with different latent-space sizes $L=\text{dim}(\mathbf{c})$ resulting in different architectures with different numbers of parameters, we believe it still makes sense to investigate/plot the relationship between $L$ and E by simply changing $L=\text{dim}(\mathbf{c})$.
> >
> >
> > > “[The explicitness quantity is] quite appealing intuitively, but I wonder what this quantity currently predicts, i.e. what it is a model of, what other quantities it correlates with and/or predicts”
> >
> > See our response above to the second point above under Weaknesses, where we provide new experiments that illustrate the strong correlation between explicitness and downstream performance with low-capacity probes.

---

> ### Author Response · Authors · 2022-11-26
> **Reviewer R81A: Has our response addressed your concerns?**
>
> Dear Reviewer R81A,
>
> We would like to kindly ask if our response has addressed your concerns. If so, we ask that you consider adjusting your score. If not, we are happy to provide further clarifications. Thank you again for your time and continued engagement.
>
> Best regards,
>
> Authors

---

> > ### Comment · Reviewer_R81A · 2022-12-05
> > **Update**
> >
> > Dear authors, I apologize for my lack of engagement, I was unable to reply earlier.
> >
> > The additional results and explanations make a lot of sense, and make the proposed method more appealing from my point of view. I will update my score.

---

> > > ### Author Response · Authors · 2022-12-06
> > > **Thank you**
> > >
> > > Dear Reviewer R81A,
> > >
> > > Thank you for engaging with our response and updating your review + score. We are glad to read that our additional experiments and explanations have addressed your concerns and made the method more appealing.
> > >
> > > Best regards,
> > >
> > > Authors

---

### Author Response · Authors · 2022-11-18
**General response to all reviewers and the AC**

We sincerely thank all reviewers for their time and thoughtful feedback.

We were pleased to read that the paper is *well written* (R81A, WGrh, wQrr, xBj7) and that our proposed measures are *“intuitively appealing”* (R81A), *“touch[es] an important open problem”* (R81A), and are *“interesting for disentanglement evaluation”* (WGrh).

We were also pleased to read positive statements about the **impact**—*“both the deep learning community and the (more theoretically oriented) nonlinear ICA community will benefit from the connections made in this work”* (wQrr)—and the **main idea**—*“I very much agree with the authors that one should take into account the capacity of a probe when evaluating a representation and that exploring different classes of probes with different capacities gives a more complete picture of the learned representation”* (wQrr).

The main concerns surrounded:
1. **The practical usefulness of our explicitness measure** (R81A, wQrr).
2. **Mathematical subtleties/rigor, mostly surrounding Corollary 3.4** (wQrr).
3. **The original DCI framework**, namely its apparent capacity-dependency (WGrh) and some apparent drawbacks (xBj7).

In response, we have:
1. **Added experiments which demonstrate the practical usefulness of our explicitness measure** by showing a strong correlation between explicitness and downstream performance with low-capacity probes (see our response to Reviewer R81A and Section 6.3 of the updated manuscript for details).
2. **Corrected Corollary 3.4 and its proof** (see Appendix A).
3. **Clarified points about the original DCI framework** in our responses to reviewers WGrh and xBj7, namely that it does not measure required-capacity and that the suggested drawbacks are false/incorrect.

We believe that these additional experiments and corrections have improved the paper, while the clarifications have resolved some misunderstandings about the original DCI framework and how/why we’ve extended it. Any changes and new material are highlighted in yellow in the revised manuscript. If the reviewers have any further questions or comments, we are very happy to respond during the discussion phase.

---

### Decision · Program_Chairs · 2023-01-20

**Decision:**

Accept: poster

**Justification For Why Not Higher Score:**

The paper proposes an interesting concept, but there still appear some rough edges in the execution and evaluation of the new concept.

**Justification For Why Not Lower Score:**

The paper proposed a quite novel concept.

**Metareview: Summary, Strengths And Weaknesses:**

The paper extends the DCI-framework and proposed a novel concept of explicitness (and size). Explicitness describes how accessible the true latents are from a code. They also provided an explicitness score as a metric for explicitness, which is the area under the curve of informativeness.

strength:

+ The concept of explicitness is novel and interesting.

weakness:

+ While explicitness is an interesting concept, the current way of computing E score appears to be not as principled and involve some arbitrary choices. Further, it remains a worthwhile question whether we should evaluate the explicitness using a single number.

+ Readers would appreciate a more extensive discussion of limitations of the current proposal.


**Note From Pc:**

if the above contains the word "oral" or "spotlight" please see: "oral" presentation means -> notable-top-5% and "spotlight" means -> notable-top-25%. As stated in our emails, we are disassociating presentation type from AC recommendations

**Summary Of Ac-Reviewer Meeting:**

The paper is well-written and clear. The proposal is novel and interesting. But the reviewers agree that there is some arbitrariness of how e score is being computed. Further whether we should aggregate the evaluation of e-score into a single number may not be a good idea.

There were some results about connections to identifiability. Reviewers agree that they are not particularly strong or useful but it is good to have them written down somewhere in the literature.

One reviewer brought up computation concerns, but not all reviewers agree it is a high-priority concern since it is an evaluation metric.

Reviewers agree that the paper should have a more extensive discussion of limitations.